# Tailored interventions into broad attitude networks towards the COVID-19 pandemic

**Monique Chambon**[1,2]*, **Jonas Dalege**[3], **Lourens J. Waldorp**[2], **Han L. J. Van der Maas**[2], **Denny Borsboom**[2], **Frenk van Harreveld**[1,2]

**1** National Institute for Public Health and the Environment (RIVM), Bilthoven, The Netherlands, **2** Department of Psychology, University of Amsterdam, Amsterdam, The Netherlands, **3** Santa Fe Institute, Santa Fe, New Mexico, United States of America

* monique.chambon@rivm.nl

**Data Availability Statement:** The data is posted on OSF https://osf.io/bhuvw/.

**Funding:** This research was funded by The Dutch Research Council (NWO grant 440.20.019). J.D.'s work was supported by an EU Horizon 2020 Marie

## Abstract

This study examines how broad attitude networks are affected by tailored interventions aimed at variables selected based on their connectiveness with other variables. We first computed a broad attitude network based on a large-scale cross-sectional COVID-19 survey ($N$ = 6,093). Over a period of approximately 10 weeks, participants were invited five times to complete this survey, with the third and fifth wave including interventions aimed at manipulating specific variables in the broad COVID-19 attitude network. Results suggest that targeted interventions that yield relatively strong effects on variables central to a broad attitude network have downstream effects on connected variables, which can be partially explained by the variables the interventions were aimed at. We conclude that broad attitude network structures can reveal important relations between variables that can help to design new interventions.

## Introduction

The persistent reluctance of many people to adopt recommended preventive behaviors during the COVID-19 pandemic has illustrated how difficult effectively promoting behavioral change can be, especially when the behavior is determined by a complex interplay of factors. The pandemic provided a unique opportunity to empirically study interventions among the public in relation to an inherently complex subject matter. An approach that is increasingly adopted when studying the complex interplay of psychological factors and effects of interventions is the network perspective. In this study, we empirically explore how the network perspective can inform social psychological interventions. The aim of the current research is to investigate how broad attitude networks respond to tailored interventions aimed at variables that differ in their connectiveness with other variables.

### Network perspective

Psychological research taking a network perspective was first used as a theoretical model in cognitive developmental research [1, 2]. It was later developed into a psychometric model,

Curie Global Fellowship (no. 889682). The funders had no role in study design, data collection and analysis, decision to publish, or preparation of the manuscript.

**Competing interests:** The authors have declared that no competing interests exist.

extensively applied in clinical psychology [3]. This perspective has recently also been adopted in the study of attitudes. The Causal Attitude Network (CAN) model [4] conceptualizes attitudes as networks, consisting of evaluative reactions (nodes) and interactions between them (edges, i.e., links between the nodes). Edges represent either excitatory or inhibitory relations with varying weights (i.e., strength of relations varies between evaluative reactions). These nodes, consisting of the cognitive, affective and behavioral elements of attitudes, together with the edges connecting them, form a network. To exemplify, Fig 1 shows a hypothetical and simplified network concerning attitudes towards hand hygiene during pandemics. Dalege, Borsboom [5] show how the CAN model explains individual attitudes and their dynamics. In Dalege and van der Maas [6] this model is used to explain the differences between implicit and explicit attitude measures.

Networks can be calculated with (e.g., survey) data. Nodes represent the measured psychological constructs, which can consist of single items or a combination of items (e.g., average on multiple items). Edges between nodes in psychological networks cannot be directly observed and are therefore parameters that are estimated from data [7]. In Gaussian networks estimated with continuous and ordinal data, edges represent partial correlations between nodes, where the correlation between two nodes is computed conditional on all other nodes in the network [e.g., see 8, 9]. Attitude network models can also be statistically estimated with empirical data

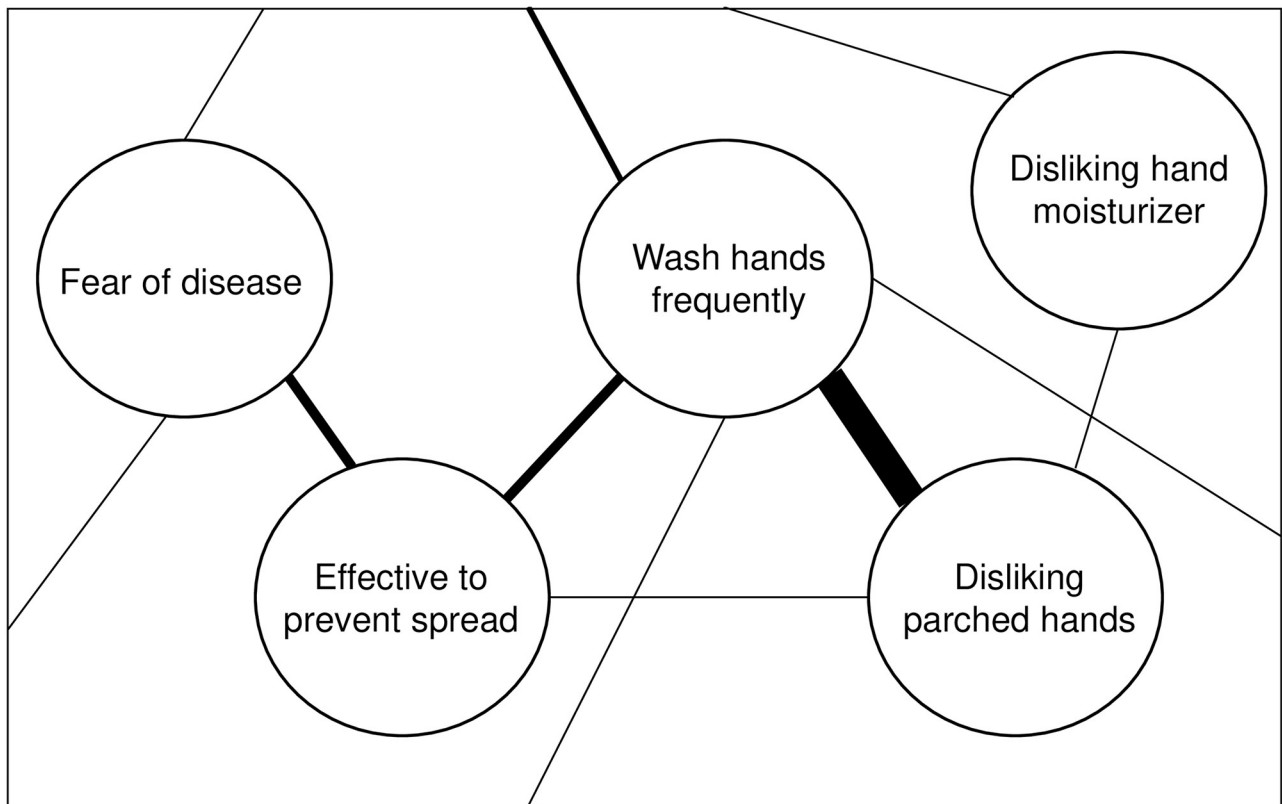

**Fig 1. Part of hypothetical and simplified attitude network.** The network concerns the attitude towards hand hygiene, consisting of a behavioral element ('Washing hands frequently'), a cognitive evaluation ('Hand hygiene is effective to prevent spread') and three affective evaluations ('Fear of disease' and 'I dislike [parched hands resulting from hand hygiene / hand moisturizer]'). Strength of the relations is indicated by edge width. In this example, disliking parched hands has a stronger relation to washing hands frequently than perceiving hand hygiene as effective to prevent the spread, indicating that the former consideration has a stronger association with the behavioral element than the latter (which is indicated in the network by different edge width). Furthermore, although disliking hand moisturizer is indirectly associated with the behavioral element of washing hands through the disliking parched hands node, this affective evaluation is (in this fictitious case) not directly related to washing hands.

[10]. Empirical applications of the CAN model are predominantly based on cross-sectional data [e.g., 11–13]. This enables computing undirected networks: networks in which edges represent associations and do not provide information on directions of these relations. Calculating networks with *directed* edges that represent predictive effects is also possible but requires a repeated measures design.

In addition to graphically representing the interplay of variables, network analysis also computes valuable properties of networks such as overall connectivity (i.e., average strength of connections between nodes) and centrality of nodes within the network (with central nodes having more and/or stronger connections to other nodes). The most commonly used centrality measures are Strength (calculated by summing absolute edge weights), Closeness (calculating distances between nodes based on the shortest path length) and Betweenness (calculated based on how often it lies on the shortest path between nodes) of nodes [14]. Of these centrality metrics, Closeness and Betweenness are considered least suitable for psychological networks [15], and can be problematic because they treat associations between nodes as distances [14].

In the context of attitudes, connectivity of an attitude network determines its stability and resistance to change [16], while centrality informs about a node's connectedness with other nodes in the attitude network. Such network properties are also considered informative for designing effective interventions. Theoretically, changing a central (i.e., highly connected) node is likely to have a more profound effect on a network than changing a peripheral (i.e., less connected) node, given the central nodes' relatively high connectiveness to other nodes [17]. This is however not undisputed in undirected networks for reasons such as missing latent common causes and problems regarding the specification of boundaries of networks [see 15, 18, 19 for an overview]. Also, high connectivity of a node can result from different scenario's, namely a) that the node highly affects other nodes, b) that the node is highly affected by other nodes, or c) a combination of the first two directions of effects. Results of interventions depend on these directions of effects: intervening on a node that highly affects other nodes is likely to have a profound effect on the network, whereas changing a node that is highly affected by other nodes is unlikely to (durably) affect the network. Nevertheless, some studies have demonstrated the value of a network perspective in studying interventions [20–22].

One example of applying the CAN model to interventions is by Wicker, Nohlen [22], in the context of sustainable consumer behaviors. They determined consumers' attitude network about plastic and included their willingness to pay for sustainable alternatives as a node. It was found that a persuasive message aimed at a central node that was most strongly connected to this willingness to pay was an effective means to change actual behavior. This underlines that interventions tailored to attitude networks can be effective in changing behavior [22]. This study specifically focused on behavior, and research into how interventions impact attitude networks as a whole seems to be mostly lacking. Two notable exceptions were conducted in the mental health domain. Bekhuis, Schoevers [20] and Blanken, Van Der Zweerde [21] demonstrated how a network approach can be used to gain insight in treatment effects aimed at specific symptoms of depression and insomnia. By including the treatment in the network as a node they provide insight into direct and indirect treatment effects.

Thus, although decades of research has investigated how attitudes are affected by interventions [23–25], research into attitude networks and change is sparse. Such research could however improve our understanding of (indirect) intervention effects. In addition to the effects of interventions on the node the intervention was aimed at, there can be 'downstream' effects (on other nodes) in the attitude network. This is important, since certain structures can either reinforce or attenuate effects of interventions, depending on connections between nodes. For instance, a triangular motif (three interconnected nodes) of negative relations between nodes

can possibly undermine intervention effects: an intervention aimed at increasing node A can lead to decreasing node B, which can lead to increasing node C, which can lead to decreasing node A, thus undermining the intervention's effect. This depends not only on the type of connections (i.e., positive or negative), but also on the number and weights of connections with other nodes (i.e., centrality).

Finally, while networks and their properties are empirically determined, the question what does and what does not fall within the scope of a network is to a large extent dependent on what is included in the analysis (as in any statistical analyses). The CAN model focuses on elements that make up individual attitudes, but attitudes do not exist within a vacuum and are related to each other and to other psychological variables. Accordingly, the scope of attitude networks has recently been broadened by including attitudes as well as a wider range of social psychological constructs such as trust and social norms. This broad attitude networks approach provides insight into the interplay of attitudes and their elements within a system of other relevant variables [12]. In the present research we also focus on such broad networks, encompassing different attitudes as well as other psychological variables. With this we aim to do justice to the inherent complexity of health behavior in the context of a pandemic, in this case the COVID-19 pandemic.

## The present research

We aim to contribute to designing effective interventions in complex psychological systems by providing empirical insight into how network structures explain intervention effects. In this study, we examine how broad attitude networks respond to tailored interventions aimed at specific nodes selected based on network properties (i.e., connectiveness with other nodes). We investigate whether these interventions a) affect the node the intervention was aimed at (i.e., target node), b) affect nodes other than target nodes, and if so, whether these effects can be explained by connections with target nodes (i.e., downstream effects). It is hypothesized that interventions impact target nodes and that node centrality predicts whether interventions have downstream effects on other nodes as well. We expected that changing more central nodes affected broad attitude networks more than changing peripheral nodes, because central nodes are more connected to other nodes in the network. Furthermore, we expect that downstream effects of interventions are induced through the nodes that interventions were aimed at, thus that target nodes explain effects of interventions beyond the target node. Note that the wording of affecting *nodes* is used for brevity and refers to affecting *scores* on the item(s) forming that node. States and changes of nodes were only measured at the behavioral level (i.e., observed responses on survey items). Our research can therefore only speak to that level of explanation and is not on the level of mental processes. The current study thus goes beyond what earlier intervention studies have done [e.g., 22], by formally comparing intervention effects on nodes with relatively high and low centrality, and testing mediation effects.

To examine the above, we designed a longitudinal study in which participants responded to a survey on COVID-19 related variables five times over a period of approximately 10 weeks (April 23rd–June 30th 2020). This study design also enables examination of the temporal dynamics of the broad COVID-19 attitude network. This is however not the focus of the current paper and will be addressed in the discussion section. The COVID-19 pandemic was deemed a suitable empirical setting to conduct this study, given the pandemic's complex and unprecedented nature, and the importance of (preventive) behavior. The psychological variables included in the survey are based on an extensive literature review of health behavior during a pandemic by Bish and Michie [26]. Their proposed frameworks, with determinants depending on types of protective behaviors, are covered by several prominent generic models

within psychology contributing to explaining behavior, such as the Theory of Planned Behavior [TPB; 27] and Health Belief Model [HBM; 28], but also include other social psychological factors relevant in the context of pandemics. These factors, for instance trust, perceived knowledge, health complaints, well-being and individual differences, were also identified in more recent literature on psychological determinants of compliance during pandemics (see S1.1 in S1 File for more information and literature). Importantly, this research does not aim for an exhaustive combination or comprehensive network of variables relevant for compliance, since such a set of variables is arguably extremely large. Instead, it aims to demonstrate how networks with variables relevant for compliance during pandemics that extend beyond attitudes, respond to tailored interventions. We present the current research in two parts: the first part presents the case study of this research (i.e., broad COVID-19 attitude network), and the second part presents results of intervening in the broad COVID-19 attitude network.

## Materials and methods

### Participants and design

This study was approved by the Ethics Review Board of the University of Amsterdam (2020-SP-12194) and not preregistered due to its explorative nature. Dutch participants were recruited via a research panel from Ipsos. Regarding sample size, the aim was to collect as large a sample as possible to ensure sufficient power to find between-subject effects in the last measurement, after which we checked the stability of the estimated network. We aimed for, and far exceeded, a minimum of 500 participants because this is the highest advised number of participants for a moderately sized network with either continuous or binary data [9, 29], and these types of data were combined in the current study. These sample sizes are advised in order to obtain accurate network estimation, indicating that the estimated network is an accurate representation of the true underlying network [7]. The initial sample was representative of the Dutch population in terms of gender, age and country region. These participants were invited for subsequent measures and no new participants were added in subsequent measures. Each measurement contained two attention checks to enhance data quality, and participants who failed both attention checks within one measurement were excluded from thereon (see Table 1).

The broad COVID-19 attitude network was based on all participants that completed the first survey (valid $N$ = 6,093). Research into effects of interventions on the network was based on two samples: the first sample included respondents that participated in the third wave in which we presented the first intervention, and the second sample included respondents that participated in the fifth wave in which we presented the second intervention. Respondents that participated in the first intervention (third wave) were also eligible for inclusion in the second intervention (fifth wave). Therefore, respondents that were included in the second intervention were also included in the first intervention. Table 1 provides sample and descriptive information for all samples, including waves in between.

In waves 3 and 5, interventions were designed for two nodes, with two experimental conditions per node (low / high). In the low (high) experimental conditions, interventions were aimed at decreasing (increasing) scores on nodes. Each wave also contained a control condition, resulting in a total of five intervention conditions for both wave 3 and 5. Interventions in wave 3 were aimed at the nodes *Trust* and *Social Norm*. Wave 5 contained interventions aimed at the nodes *Measures Support* and *Economic Consequences*. Again, these nodes were selected based on network properties, and therefore not necessary directly related to the behavioral element (i.e., compliance) in the broad COVID-19 attitude network. More specifically, target nodes were selected based on different node strength (i.e., central and peripheral node) in the

**Table 1. Demographic and intervention specifics.**

| | Measure | Wave 1 | Wave 2 | Wave 3 | Wave 4 | Wave 5 |
|---|---|---|---|---|---|---|
| **Sample formation** | | | | | | |
| **Start data collection (2020)** | | 23[th] April | 13[th] May | 27[th] May | 10[th] June | 24[th] June |
| **End data collection (2020)** | | 5[th] May | 18[th] May | 2[th] June | 16[th] June | 30[th] June |
| **Failed attention check** | $n$ | 519 | 118 | 41 | 24 | 8 |
| **Sample (including missing values)** | $n$ | 6,219 | 4,953 | 3,754 | 2,822 | 2,449 |
| **Drop-out**[a] | $n$ (%) | | 1,266 (20.4%) | 1,199 (24.2%) | 932 (24.8%) | 373 (11.4%) |
| **Missing values**[b] | $n$ (%) | 126 (2.0%) | 89 (1.0%) | 70 (1.9%) | 58 (2.1%) | 50 (2.0%) |
| **Valid sample** | $N$ | 6,093 | 4,864 | 3,684 | 2,764 | 2,399 |
| **Demographics Valid $N$** | | | | | | |
| **Gender** | % female | 51.4% | 50.6% | 50.1% | 49.1% | 49.4% |
| **Age** | Range (years) | 16–89 | 16–89 | 18–89 | 18–89 | 18–89 |
| | $M$ (SD) | 49.32 (16.72) | 51.20 (16.32) | 51.99 (16.30) | 53.42 (16.06) | 53.69 (15.84) |
| **Education** | % primary or secondary education | 55.3% | | | | 54.6% |
| | % higher education | 44.7% | | | | 45.4% |
| **Illness** | % confirmed | 30.6% | | | | 34.9% |
| **Smoking** | % confirmed | 17% | | | | 15.9% |
| **Interventions** | $n$ (%) passed manipulation checks | | | Total $n$ = 2,845 (77.2%) | | Total $n$ = 2,123 (88.5%) |
| | Per intervention condition | | | Control T3 ($n$ = 702, 96.0%) | | Control T5 ($n$ = 445, 94.1%) |
| | | | | Trust (low: $n$ = 536, 72.3%; high: $n$ = 507, 69.5%) | | Measures Support (low: $n$ = 378, 80.3%; high: $n$ = 456, 94.2%) |
| | | | | Social Norm (low: $n$ = 689, 92.7%; high: $n$ = 411, 55.5%[c]) | | Economic Consequences (low: $n$ = 401, 82.7%; high: $n$ = 443, 91.2%) |

[a] Formal comparison of the network structure of the longitudinal sample and drop–outs revealed only two significantly different edges. More information on participants that dropped out of the longitudinal study is provided in S1 File (S2.1).

[b] Education, illness and smoking included the answer 'I prefer not to answer', which was treated as a missing value. Participants with missing values for one or more nodes were deleted from analysis due to the small number of missing values and the methodological challenges pairwise deletion would impose on network analysis.

[c] The percentage of participants passing the manipulation check of the high social norm condition was rather low, possibly because messages about decreasing adherence in the public dominated the news during data collection in Wave 3.

first wave and feasibility of designing a targeted intervention (see results section for more information on selecting nodes). As mentioned, node strength is considered the most suitable centrality measure for psychological networks. This research thus focuses on node strength as a centrality measure. Node strength represents the conditional association between a node and other nodes in the network and is meant to aid interpretation of networks. It is calculated by summing absolute weights of edges a node has with connected nodes. High node strength thus represents the number and strength of a node's relations but does not inform us about directions of relations: edges can represent effects from or to nodes, or be bidirectional.

The interventions contained a manipulation check consisting of a multiple-choice question following the intervention to assess whether participants read provided information (e.g., Social Norm conditions: 'According to the information above, has the level of compliance with

the corona measures increased, decreased or remained the same?'). Table 1 reports the number and percentage of participants answering this question correctly, thus passing the manipulation check, resulting in the intervention subsamples.

## Measures

Data was collected with an online survey in Dutch. The first step in composing the survey was to identify relevant constructs in the literature (see S1.1 in S1 File). As mentioned, our aim was not to include an exhaustive set of variables related to behavior during pandemics, nor to evaluate a specific (set of) model(s). Instead, we identified important constructs that extend beyond attitudes to include in the network to broaden the scope of the network. Subsequently, a survey was developed with items based on these constructs. An overview of the survey items is provided in S1 File (S1.2). After data collection in wave 1, we constructed psychological variables as nodes by either a predetermined combination of items or based on components in the data as identified through Principal Axis Factoring (PAF; a data reduction technique). Constructs that were surveyed with a single or two items were predetermined nodes (see Table 2 for the number of items per node). Other predetermined nodes were validated scales (see Table 2 –the nodes with references in the footnotes were the adopted validated scales). The remaining nodes were constructed based on the results of the PAF. S1 File (S2.2) provides node-specific descriptions of the approach to combine items into nodes, including PAF results.

Table 2 presents the resulting nodes, including examples of survey items for those constructs and their answer scales. Nodes that were based on multiple items consisted of mean scores of items relevant for that node, except for *Risk Perception* (i.e., the product of likelihood and severity of an infection) and *Mental Well-being* (i.e., sum score). A detailed description of each node in the network, its interpretation and the scale reliability as observed in the current study is provided in S1 File (S2.3).

## Procedure

Participants that subscribed to Ipsos' research panel received an invitation via e-mail to participate in our study. Only participants that finished the survey received an invitation for subsequent waves. They received compensation in the form of points that can be spent at web shops. Participants were informed about participation and provided written consent. They were also asked to commit to the longitudinal research design beforehand.

Participants were randomly assigned to an intervention condition by the software in which the questionnaire was programmed (i.e., Qualtrics). The intervention was presented at the beginning of the survey within a specific wave. Each experimental condition consisted of two headings of online news articles and additional information in text, followed by a question that served as a manipulation check. Additional information consisted of (fictional) preliminary results of the current study supporting the presented news articles and possible explanations for these results. The control condition consisted of one news article and a manipulation check. The interventions, including references to the news articles, can be found in S1 File (S1.3). Fig 2 shows an example of an intervention (high social norm condition; original in Dutch), which argued that the vast majority of Dutch people follow and support behavioral guidelines [i.e., descriptive and injunctive norm; 38]. The control condition in wave 3 and wave 5 covered news on topics that were expected to minimally affect nodes: increases in Netflix-subscribers in wave 3 and increases in sale of used cars in wave 5.

**Table 2. Nodes (psychological variables) based on items in the survey, including item examples and answer scales.**

| Node (items per node) | Examples of items per node (/ in the same text line means separate item in survey) | Scale |
|---|---|---|
| **Compliance (5)** | Keep 1.5 meters away from others. / Wash your hands regularly with water and soap. / Cough and sneeze into the inside of your elbow. | 1 (*I do not display this behavior more*) to 7 (*I display this behavior much more now*) |
| **Risk Perception (2)** | How likely (/ severe) do you believe it is you will get infected with the coronavirus within the next year? | 1 (*Extremely unlikely*) to 7 (*Extremely likely*) |
| **Health Risk (2)** | For me personally (/ my family and friends), I consider the health risk of an infection with the coronavirus. . . | 1 (*Extremely small*) to 7 (*Extremely severe*) |
| **Economic Consequences (2)** | For me personally (/ my family and friends), I consider the economic consequences of the corona pandemic. . . | 1 (*Extremely small*) to 7 (*Extremely severe*) |
| **Self-exempting Beliefs (2)** | I will not get infected with the coronavirus because I never get the seasonal flu either. / I think I am already immune (protected) against the coronavirus. | 1 (*Strongly disagree*) to 7 (*Strongly agree*) |
| **Negative Affect (8)** | The corona pandemic is making me (feel). . . (e.g., angry / sad / confused / uncertain) | 1 (*Strongly disagree*) to 7 (*Strongly agree*) |
| **Compassion (1)** | The corona pandemic is making me feel compassion. | 1 (*Strongly disagree*) to 7 (*Strongly agree*) |
| **Worries Virus^ (4)** | I worry about.. (e.g., getting infected / losing someone I love / the health care system overloading) | 1 (*Do not worry at all*) to 7 (*Worry a lot*) |
| **Worries Measures^ (6)** | I worry about.. (e.g., what staying at home a lot will do to my health / a recession / getting lonely) | 1 (*Do not worry at all*) to 7 (*Worry a lot*) |
| **Vaccination Intention^ (1)** | If a vaccine becomes available, I would get it. | 1 (*Strongly disagree*) to 7 (*Strongly agree*) |
| **Measures Support (7)** | I find the corona measures.. (Senseless-Sensible / Useless-Useful / Unnecessary-Necessary) | 1 (Negative option) to 7 (Positive option) |
| **Measures Ease (2)** | I find the corona measures.. (Unpleasant–Pleasant / Difficult-Easy) | 1 (Negative option) to 7 (Positive option) |
| **Social Norm (2)** | I think the majority of people (/ find it important that people) comply with the corona measures. | 1 (*Strongly disagree*) to 7 (*Strongly agree*) |
| **Control Infection (2)** | For me personally (/my family and friends), avoiding an infection with the coronavirus in the current situation is.. | 1 (*Extremely difficult*) to 7 (*Extremely easy*) |
| **Self-efficacy (1)** | I know how to protect myself from the coronavirus. | 1 (*Strongly disagree*) to 7 (*Strongly agree*) |
| **Involvement (3)** | To what extent does the news about the corona pandemic have your attention? / How much do you think about the corona pandemic? | 1 (*Not at all*) to 7 (*Very much*) |
| **Perceived Knowledge (1)** | How much knowledge do you think you have about the corona pandemic? | 1 (*Very little*) to 7 (*Very much*) |
| **Trust (4)** | I trust.. (e.g., the authorities to adequately manage / health care professionals during).. the corona pandemic. | 1 (*Strongly disagree*) to 7 (*Strongly agree*) |

(*Continued*)

**Table 2.** (Continued)

| Node (items per node) | Examples of items per node (/ in the same text line means separate item in survey) | Scale |
|---|---|---|
| **Consideration of Future Consequences[a] (5)** | I am willing to sacrifice my immediate happiness or well-being in order to achieve future outcomes. / I think it is important to take warnings about negative outcomes seriously even if the negative outcome will not occur for many years. | 1 (*Strongly disagree*) to 7 (*Strongly agree*) |
| **Resilience[b] (6)** | I tend to bounce back quickly after hard times. / I usually come through difficult times with little trouble. | 1 (*Strongly disagree*) to 5 (*Strongly agree*) |
| **Coping[c] (10)** | I think that I have to accept that this has happened. / I think about a plan of what I can do best. | 1 (*[almost] Never*) to 5 (*[almost] Always*) |
| **General Health (1)** | In general, how would you rate your health? | 1 (*Very poor*) to 7 (*Very good*) |
| **Health change Physical (1)** | How would you rate your physical health now as compared to before the corona pandemic? | -3 (*Much worse*) to 3 (*Much better*) |
| **Health change Mental (1)** | How would you rate your mental health now as compared to before the corona pandemic? | -3 (*Much worse*) to 3 (*Much better*) |
| **Healthy Lifestyle (3)** | I've been (eating / exercising / sleeping) in the past two weeks, compared to before the corona pandemic. . . | -3 (*Much [less healthy / less / worse]*) to 3 (*Much [healthier / more / better]*) |
| **Mental Well-being[d] (7)** | I've been feeling optimistic about the future/ I've been feeling relaxed. / I've been thinking clearly. | 1 (*Never*) to 5 (*Always*) |
| **Loneliness[e] (6)** | I experience a general sense of emptiness. / I miss having people around me. | 1 (*Not at all*) to 5 (*Very much*) |
| **Complaints[f] (somatic [6]; depressive [6]; anxiety [6])** | To what extent did you experience.. (e.g., faintness / chest pain / nausea; e.g., loss of interest / worthlessness / loss of the will to live; e.g., nervousness / restlessness / panic attacks) during the past two weeks? | 1 (*Not at all*) to 5 (*Very much*) |
| **Illness (1) / Smoke (1) / Age (1) / Gender (1) / Education (1)** | Do you suffer from one or more of the following conditions? (e.g., cancer, seriously overweight) / Do you smoke? / How old are you? / What is your gender? / What is your highest level of education? | 0 (*No*) to 1 (*Yes*) / *n/a* / Open numeric field / 0 (*Male*) to 1 (*Female*) / 0 (*Primary/ secondary*) to 1 (*Higher*) / n/a |

The sections of the survey that referred to 'the corona measures' contained the following explanatory text 'By this we mean the recommendations to prevent the spread of the coronavirus and thus prevent overloading the healthcare system, for example stay at home as much as possible, keep 1.5 meters of distance from others and wash your hands regularly with soap and water.'

^ WHO Regional Office for Europe [30], Multiple items (e.g., worries, vaccination intention) were adopted from the WHO protocol for COVID–19 monitoring.

[a] Strathman, Gleicher [31], answer scale formally ranges from 1 (*Extremely uncharacteristic*) to 5 (*Extremely characteristic*).

[b] Smith, Dalen [32].

[c] Garnefski and Kraaij [33], the following subscales were adopted based on Kalisch, Veer [34]: Acceptance, Positive refocusing, Refocus on planning, Positive reappraisal and Putting into perspective.

[d] Tennant, Hiller [35], raw scores were converted to metric scores as required for the (S)WEMWBS.

[e] de Jong Gierveld and van Tilburg [36], answer scale formally ranges from 1 (No!) to 5 (Yes!).

[f] Derogatis [37], answer scale formally ranges from 0 (*Not at all*) to 4 (*Extremely*).

Before we continue to the third survey, we would like to share a couple of news articles with you.

*The current study*
Despite the fact that some places are getting crowded, research shows that the vast majority of people still comply with the rules. The current study also shows that people increasingly adhere to the corona measures. The extent to which people find it important that others adhere to the corona measures has also increased.

*Explanation*
The fact that people adhere more to the corona measures is because **an increasing amount of people believes that the measures are proportional to the current phase of the corona crisis**.
Slightly looser where possible, but careful, to avoid later regrets, as Prime Minister Rutte says.

| **99 percent of the Dutch say they keep a distance of 1.5 meters** | **The Dutch stay at home despite beautiful spring weather** |
|---|---|
| Picture of sign with the text 'keep distance' | Picture of two people walking outside in nature |
| **Almost all Dutch people adhere to the behavioral rules that must prevent infection with the coronavirus. 99 percent says they keep the requested 1.5 meter distance from others, 97 percent wash their hands more often and 93 percent stays at home as much as possible. More than half addresses others about violating these rules.** | **Despite the beautiful spring weather, the Netherlands seems to be following the advice to stay at home as much as possible. According to various regions and authorities, it has been quiet today in the nature reserves.** |

**Fig 2. Example of intervention for 'Social Norm high' condition.** News articles are replaced due to copyright. Sources of original news articles are provided in S1 File (S1.3).

## Analysis

The broad COVID-19 attitude network was estimated using *mgm* [Mixed Graphical Models to combine continuous and binary variables; 39] with k = 2 (all pairwise interactions). An edge was included in the network if any of the two possible directions between edges were selected with 10-fold cross-validation (lambdaSel = CV [cross validation] and lambdaFolds = 10, both default settings of mgm). We have opted for k-fold CV because it emphasizes prediction more than with the EBIC, but often they give similar results. Both methods have also been investigated and have good properties. The k-fold CV is sometimes more conservative (i.e., allows fewer edges) than the EBIC, especially for smaller sample sizes. Besides the values specified in the R-script (see S2.4 in S1 File), the default values were used. Readers are referred to Haslbeck and Waldorp [39] for in-depth information about this method. Communities (i.e., groups of highly interconnected nodes) were identified with a walktrap algorithm (see S3.1.2 in S1 File).

One-way ANOVA and Kruskal-Wallis tests were conducted to examine differences between intervention conditions (i.e., low, high and control condition). The Kolmogorov-Smirnov value was significant for all variables ($p < .001$). Based on the QQ-plots, the variables *Health Risk*, *Economic Consequences*, *Negative Affect*, *Worries Virus* and *Measures Ease* were treated as normally distributed and therefore analyzed with a one-way ANOVA test. The rest of the variables were analyzed with the non-parametric Kruskal-Wallis test.

Given the many effects of the interventions, we adjusted the alpha level to $p < .01$ to reduce the type I error probability (see S3.2.2 in S1 File for significance values of each node per intervention). This adjusted significance value is determined based on the fact that we hypothesized a number of effects based on edges in the network, so Bonferroni correction for multiple comparisons based on all 27 nodes would increase type II errors, and tailored significance levels per intervention would decrease readability. Note that this adjusted significance level does not apply to post hoc tests given their included correction for multiple comparisons. For ANOVA, in case of equal variances, post hoc analysis was conducted with Tukey, and in case of unequal

variances, Welch statistics are reported and post hoc analysis was conducted with Games-Howell. For Kruskal-Wallis, post hoc analysis was conducted with Dunn-Bonferroni post hoc method, and significance values were adjusted with a Bonferroni correction for multiple comparisons. Mean ranks are reported for significant differences between conditions with similar medians.

Mediation analyses were conducted with PROCESS in R [40]. We conducted multi-categorical independent variable mediation analyses with the control group as the reference group [41]. Given this reference group, mediation analyses were conducted for interventions resulting in significant differences for scores on target nodes between control and experimental conditions.

## Results

The first part presents the broad COVID-19 attitude network that serves as the case study in this research, and the second part presents results of intervening in this network.

### Broad COVID-19 attitude network structure

The broad COVID-19 attitude network, obtained through nodewise regression, is shown in Fig 3a (left). Nodes represent measured psychological factors. In general terms, the right section of the network displays (psychological) health nodes. The left section is comprised of cognitive and behavioral attitude nodes and additional psychological nodes (e.g., social norm, perceived knowledge, trust and perceived control). Edges represent linear relations between two nodes after controlling for every other node in the network. These edges are associations, meaning that the direction of the relation is not determined. Interpretation guidelines can be found in the caption of Fig 3a. Edge weights are regression coefficients that represent the strength of a relation between two nodes after removing effects from all other nodes in the network. Edges with weights below the value of .10 are omitted from the figure to facilitate readability (see S3.1.3 in S1 File for network without threshold). The edges discussed in this section had sufficient estimation accuracy and their edge weights are reported in parentheses. A complete overview of the edge weights and their accuracy is provided in S1 File (S3.1). In order to provide a brief descriptive account of the broad COVID-19 attitude network, we focus on the behavioral (i.e., compliance) and health elements (see S3.1.1 in S1 File for a more detailed interpretation of the network).

Nodes in the broad COVID-19 attitude network that showed the strongest relation with *Compliance*, were *Gender* (.28), *Measures Support* (.24) and *Self-efficacy* (.13). This indicates that compliance was positively associated with being female, support for behavioral measures and perceived self-efficacy. Edges between *Compliance* and *Gender*, and *Compliance* and *Measures Support*, were of comparable weight, as indicated by edge weight and formally tested with an edge difference test (see S3.1.6 in S1 File). Furthermore, these edges were both significantly stronger than the edge between *Compliance* and *Self-efficacy*. Regarding nodes on individual differences, *Compliance* was positively related to *Consideration of Future Consequences* and *Coping* (both .07).

The relatively important nodes that were directly associated with *Mental Well-being* were *Coping* (.22; significantly strongest positive edge), *Resilience* (.16; significantly second strongest positive edge) and *Health General* (.12), but also *Loneliness* (-.19; significantly strongest negative edge) and *Depressive Complaints* (-.15). This implies that higher mental well-being during the COVID-19 pandemic was related to a positive coping strategy, more resilience, and better perceived general health, and with less experienced loneliness and depressive complaints. Two pairs of these nodes connected to *Mental Well-being* were also directly connected among each

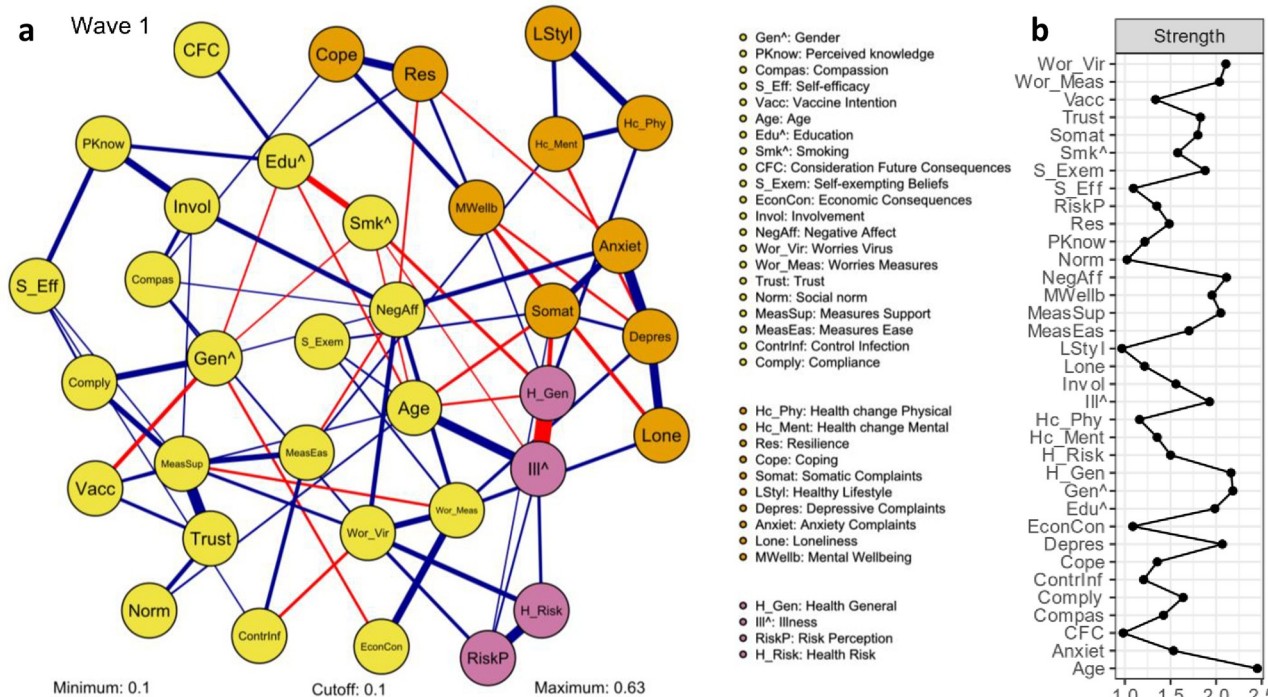

**Fig 3. Psychological broad COVID–19 attitude network.** a) Network obtained through nodewise regression, based on sample of participants that completed the first wave (*N* = 6,093). Nodes represent the measured psychological factors and edges represent the relations between nodes. For binary nodes (marked with ^), a positive relationship indicates that increasing the other node results in a higher probability for category one of the binary node (i.e., Gender 1 = Female; Education 1 = Higher; Smoking 1 = Yes; Illness 1 = Yes). Positive edges (blue) represent excitatory relations and negative edges (red) represent inhibitory relations. The strength of the relation is indicated by the edge weight (visualized by edge width). Edges with edge weights below the value of .10 are omitted to facilitate readability. Colored groups represent communities (i.e., clusters with higher interconnectedness) consisting of nodes being more connected to each other than to other nodes in the network; b) Centrality measure 'Strength' for each node in the broad COVID–19 attitude network. This measure represents the conditional association between a node with other nodes in the network and is calculated by the sum of absolute edge weights of relations a specific node has with connected nodes.

other, namely *Coping* and *Resilience* (.33), and *Depressive Complaints* and *Loneliness* (.36). Such a triangle is a particular motif that lends support to the idea that there is a reinforcing structure.

The broad COVID-19 attitude network also showed that *Vaccination Intention* was higher among men (-.20) and positively associated with *Trust* (.17) and *Age* (.12). Another salient detail was that the node *Negative Affect*, positioned at the center of the network, had eight relations to other nodes in the network: the significantly strongest, positive edges of comparable weight with *Involvement* (.23), *Anxiety Complaints* (.22), *Worries Virus* (.21) and *Worries Measures* (.20), followed by the significantly weaker edges of comparable weight with *Gender* (.11) and *Compassion* (.10), and negative edges with *Measures Ease* (-.13) and *Resilience* (-.13).

**Centrality.** Fig 3b presents the node strength measure for the psychological broad COVID-19 attitude network. The stability of this centrality measure for the network was sufficient (see S3.1.6 in S1 File). The relatively moderate strength of the node *Compliance* (1.64) in the network suggests that this node's conditional association with other nodes in the broad COVID-19 attitude network is moderate. Concerning strength of the nodes that had a direct relation with *Compliance*: the nodes *Gender* (2.18) and *Measures Support* (2.05) had the highest and comparable node strength, followed by the significantly lower node strength of *Self-efficacy* (1.09). This suggests relative high importance of *Gender* and *Measures Support* for the network due to the amount and weight of edges with other nodes.

The relatively high node strength of *Mental Well-being* (1.95) in the broad COVID-19 atti-
tude network indicates a relatively high conditional association with other nodes. Regarding
the strength of the nodes that had a direct relation with *Mental Well-being* in the network, the
nodes *Health General* (2.16) and *Depressive Complaints* (2.07) had the highest and comparable
node strength, followed by the significantly lower but mutually comparable node strengths of
*Resilience* (1.49) and *Coping* (1.35). Subsequently, the significantly lower node strength of
*Loneliness* (1.22) follows, differing significantly from *Resilience*, but not from *Coping*. This
indicates that, regarding nodes related to *Mental Well-being*, perceived general health and
depressive complaints have the relatively highest conditional association with other nodes in
the network.

Interestingly, all but one attitudinal affective node showed relatively high node strength,
indicating that these nodes were central and therefore potentially important for the broad
COVID-19 attitude network. More specifically, the nodes that were of relatively high and com-
parable strength were *Negative Affect* (2.11), *Worries Virus* (2.11) and *Worries Measures* (2.04).
This suggests that the attitudinal affective nodes have many and/or strong relations with other
nodes in the network.

## Intervening in the broad COVID-19 attitude network

This part presents whether interventions aimed at specific nodes in the broad COVID-19 atti-
tude network a) affected the targeted node (i.e., manipulation check), b) affected other nodes,
and if so, whether the network structure could explain these effects. The interventions,
included in wave 3 and 5, were aimed to affect scores of nodes that varied in node strength
(based on the results of the first wave). As mentioned, these nodes were selected based on net-
work properties, and therefore not necessary directly related to compliance in the network.
The networks of the different waves were highly stable: correlations between edge weights of
the network based on the first wave and the two waves that included interventions were $r = .96$
(wave 1 and 3) and $r = .93$ (wave 1 and 5). Table 3 presents the descriptive statistics of nodes
for the subsamples of each intervention condition in wave 3 and wave 5. These subsamples
consisted of participants that both completed the survey up to and including that wave and
passed the manipulation check (see Table 1 for the sample size of each intervention condition).
Significant results of the interventions (i.e., differences between low, high and/or control con-
ditions) are marked in Table 3 and depicted in Fig 4. A specification of these differences is pro-
vided in the text.

**Interventions wave 3.** Wave 3 included interventions aimed at the node *Trust* and *Social
Norm*, resulting in five intervention conditions to which participants were randomly assigned
(i.e., trust low, trust high, social norm low, social norm high and control condition). *Trust* was
identified as a relatively central node (node strength 1.83 in wave 1) and *Social Norm* was
selected as a peripheral node (node strength 1.02 in wave 1), with both nodes appearing to be
realistic targets for experimental manipulations (i.e., feasibility criteria).

There were significant differences between the intervention conditions aimed at the node
*Trust* for the target node *Trust*, $H(2) = 39.02$, $p < .001$, $\eta^2 = .02$, indicating successful manipu-
lation (see Table 3 and Fig 4). Post hoc analysis indicated that participants in the high trust
condition scored significantly higher on *Trust* ($Mdn = 6$) than participants in the low trust
condition ($Mdn = 5.5$, $p < .001$). Both conditions differed significantly from the control condi-
tion ($Mdn = 5.75$, $p < .001$ and $p = .028$, respectively). Based on the broad COVID-19 attitude
network structure, one could expect downstream effects of the manipulation on nodes with
the strongest edges with the target node *Trust*, namely *Measures Support* (edge weight .44),
*Social Norm* (edge weight .20) and *Vaccination Intention* (edge weight .17). Interventions

**Table 3. Statistics of nodes that differed significantly ($p < .01$) between intervention conditions in wave 3 and 5.** Including one way ANOVA/Kruskal–Wallis results for comparing conditions (including control condition).

| Nodes (7-point Likert-scale) | Wave 3 (Valid $n$ = 2,845) | | | | | Wave 5 (Valid $n$ = 2,123) | | | | |
|---|---|---|---|---|---|---|---|---|---|---|
| | Control T3 | Trust Low | Trust High | Social norm Low | Social norm High | Control T5 | Measures Support Low | Measures Support High | Economic Consequences Low | Economic Consequences High |
| | M (SD) | M (SD) | M (SD) | M (SD) | M (SD) | M (SD) | M (SD) | M (SD) | M (SD) | M (SD) |
| Negative Affect | 3.45 (1.41) | 3.42 (1.3) | 3.36 (1.36) | 3.43 (1.39) | 3.39 (1.31) | 3.07 (1.38) | 3.07 (1.45) | 3.11 (1.43) | 2.98 (1.39) | 3.28 (1.45)* |
| | Mdn (IQR) | Mdn (IQR) | Mdn (IQR) | Mdn (IQR) | Mdn (IQR) | Mdn (IQR) | Mdn (IQR) | Mdn (IQR) | Mdn (IQR) | Mdn (IQR) |
| Worries Measures | 3 (1.5) | 3 (1.33) | 3 (1.17) | 3 (1.33) | 3 (1.33) | 2.83 (1.17) | 2.67 (1.5) | 2.67 (1.5) | 2.67 (1.25) | 2.83 (1.5)* |
| Measures Support | 5.71 (1.57) | 5.57 (1.71) | 5.86 (1.43)* | 5.71 (1.57) | 5.71 (1.57) | 5.71 (1.57) | 5.71 (1.43) | 5.86 (1.57)* | 5.71 (1.57) | 5.57 (1.57) |
| Social Norm | 5 (1.5) | 4.5 (1.5) | 5 (1.5)** | 4.5 (2) | 5 (1.5)** | 4.5 (1.5) | 4 (1.5) | 4.5 (2)* | 4.5 (1.5) | 4.5 (1.5) |
| Trust | 5.75 (1.5) | 5.5 (1.25) | 6 (1)** | 5.5 (1.5) | 5.75 (1.25) | 6 (1.5) | 5.75 (1.31) | 6 (1.25) | 5.75 (1.25) | 5.5 (1.5) |

A specification of these differences is provided in the text. See S1 File (S3.2) for tables including non–significant differences and exact $p$–values.

* $p < .01$;

** $p < .001$;

indicating a significant difference between the low, high and/or control condition of the intervention–see text for specifications.

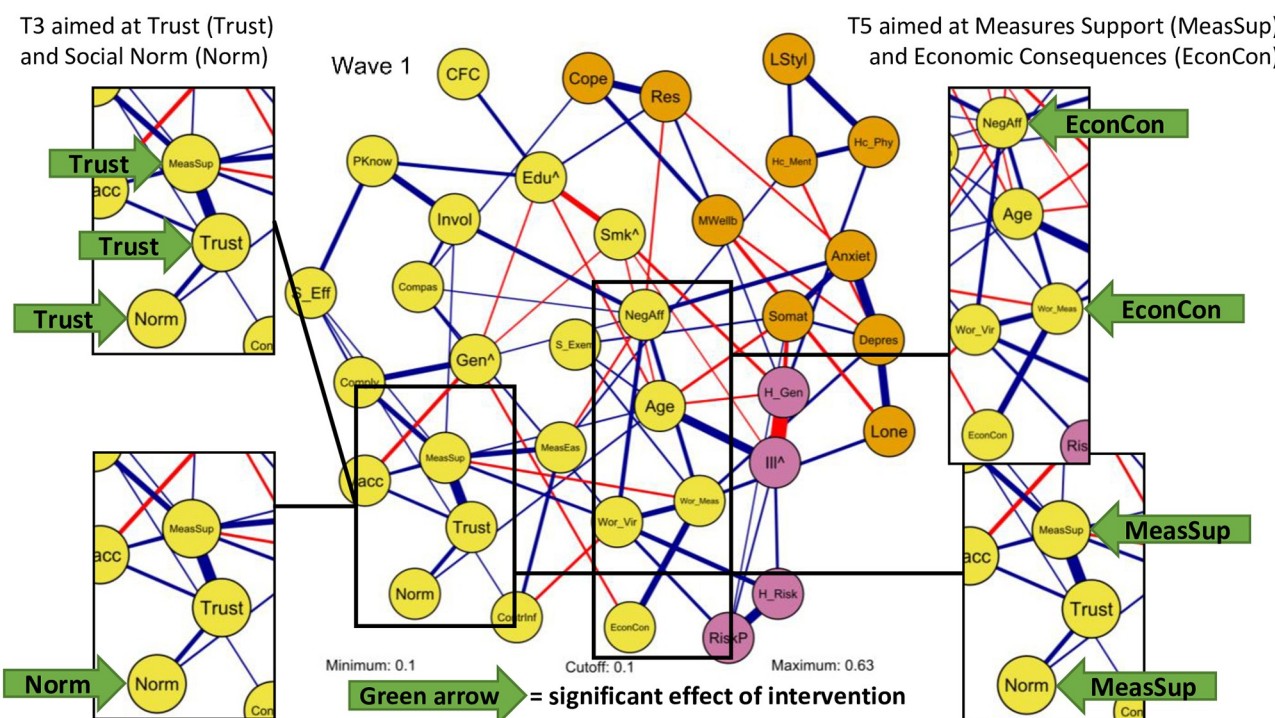

**Fig 4. Visualization of significant effects of interventions on nodes in the broad COVID–19 attitude network.** See Fig 3a for node legend. The green arrow indicates a significant effect with the specific intervention included in the arrow. Note that the broad COVID–19 attitude network is based on participants from the first wave and not the wave in which interventions were included.

indeed had significant effects on connected nodes *Measures Support*, $H(2) = 13.40$, $p = .001$, $\eta^2 = .01$, and *Social Norm*, $H(2) = 18.41$, $p < .001$, $\eta^2 = .01$. Post hoc analysis indicated that participants in the high trust condition scored significantly higher on *Measures Support* (*Mdn* = 5.86) than participants in the low trust condition (*Mdn* = 5.57, $p = .001$), whereas no significant differences were observed between the experimental conditions and control condition (*Mdn* = 5.71; control vs. low $p = .105$, control vs. high $p = .212$). Participants in the high trust condition also scored significantly higher on *Social Norm* (*Mdn* = 5, mean rank 945.92) than participants in the low trust condition (*Mdn* = 4.5, mean rank 814.43; $p < .001$) or control condition (*Mdn* = 5, mean rank 865.06; $p = .016$). There were no significant differences between the low trust condition and control condition for *Social Norm*, $p = .230$. Furthermore, there was no significant effect for the connected node *Vaccination Intention* ($p = .953$). These results suggest that there might be a causal effect from *Trust* on *Measures Support* and *Social Norm*. Note that these results do not provide information about reversed effects from those nodes to *Trust* (i.e., bidirectional relations), which might also exist.

Interventions targeted at the node *Social Norm* resulted in a significant difference between intervention conditions, $H(2) = 67.90$, $p < .001$, $\eta^2 = .04$, again indicating successful manipulation (see Table 3 and Fig 4). Post hoc analysis indicated that all intervention conditions differed significantly from each other: participants in the high social norm condition scored significantly higher on *Social Norm* (*Mdn* = 5, mean rank 1,064.09) than participants in the low social norm (*Mdn* = 4.5, mean rank 799.59; $p < .001$), and both the high and low social norm condition differed significantly from the control condition (*Mdn* = 5, mean rank 906.33; both $p < .001$). Based on the broad COVID-19 attitude network structure, one could expect an effect of interventions aimed at *Social Norm* on the connected node *Trust* (edge weight .20), whereas the rest of the edges were relatively weak. No nodes other than the target node were however affected by the intervention aimed at *Social Norm* ($p$-values ranged from .097 to .959; *Trust* $p = .191$).

*Effects of manipulation on subsequent wave*. Participants included in interventions aimed at the node *Trust* in wave 3 also scored significantly different on this target node *Trust* in the subsequent wave (T4), $H(2) = 9.64$, $p = .008$, $\eta^2 = .01$, which appears to underline the robustness of the intervention's effect. Similar to the effect in wave 3, post hoc analysis indicated that participants in the high trust condition scored significantly higher on *Trust* in wave 4 (*Mdn* = 5.75) than participants in the low trust condition in wave 3 (*Mdn* = 5.5, $p = .007$). The effect of the intervention differed from wave 3 in that there were no significant differences between the experimental conditions and control condition in wave 3 for *Trust* in wave 4 (*Mdn* = 5.5; control vs. low $p = .950$, control vs. high $p = .077$). The interventions aimed at the node *Social Norm* in wave 3 had no significant effect on nodes in wave 4.

**Interventions wave 5.**   In wave 5, interventions aimed at the nodes *Measures Support* and *Economic Consequences* were included, again resulting in five intervention conditions to which participants were randomly assigned (i.e., measures support low, measures support high, economic consequences low, economic consequences high, control condition). *Measures Support* was identified as a central node (node strength 2.05 in wave 1) and *Economic Consequences* was selected as a rather peripheral node (node strength 1.09 in wave 1), again with both nodes appearing to be realistic (i.e., feasible) targets for experimental manipulations.

As expected, there were significant differences of the manipulation aimed at support for the measures on the node *Measures Support*, $H(2) = 14.88$, $p = .001$, $\eta^2 = .01$ (see Table 3 and Fig 4). Post hoc analysis showed that participants in the high measures support condition scored significantly higher on the target node *Measures Support* (*Mdn* = 5.86) than participants in the low measures support condition (*Mdn* = 5.71), $p < .001$. There were no significant differences between the experimental conditions and control condition (*Mdn* = 5.71) for *Measures*

*Support* (control vs. low $p$ = .068, control vs. high $p$ = .311), indicating a partial successful manipulation. Based on the broad COVID-19 attitude network structure, one could expect effects on nodes with the strongest edge with the target node *Measures Support*, namely *Trust* (edge weight .44), *Measures Ease* (edge weight .25) and *Compliance* (edge weight .24). These effects were not observed (*Trust*, $p$ = .054; *Measures Ease*, $p$ = .048; *Compliance*, $p$ = .027). Regarding remaining nodes, a significant difference for the node *Social Norm* was observed, $H(2)$ = 14.48, $p$ < .001, $\eta^2$ = .01. The score on *Social Norm* for the low measures support condition (*Mdn* = 4) was significantly lower than the high social norm condition (*Mdn* = 4.5, $p$ = .001) and control condition (*Mdn* = 4.5, $p$ = .005), whereas the latter two did not differ significantly, $p$ = 1.00.

Lastly, there was no significant difference between intervention conditions aimed at the node *Economic Consequences*, $F(2, 856.68)$ = 4.22, $p$ = .015. Based on the broad COVID-19 attitude network structure, one could expect effects of interventions aimed at *Economic Consequences* on the connected node *Worries Measures* (edge weight .30), whereas the rest of the edges were relatively weak. Results showed significant differences for the connected nodes *Worries Measures*, $H(2)$ = 13.95, $p$ < .001, $\eta^2$ = .01. Post hoc analysis indicated that participants in the high economic consequences condition scored significantly higher on the node *Worries Measures* (*Mdn* = 2.83) than participants in the low economic consequences condition (*Mdn* = 2.67, $p$ = .001), whereas no significant differences were found between the experimental conditions and control condition (*Mdn* = 2.83) for *Worries Measures* (control vs. low $p$ = .230, control vs. high $p$ = .132). Regarding remaining nodes, there was a significant difference for the node *Negative Affect*, $F(2, 1286)$ = 5.05, $p$ = .007, $\eta^2$ = .01. Post hoc analysis indicated that participants in the high economic consequences condition scored significantly higher on *Negative Affect* (*M* = 3.28, *SD* = 1.45) than participants in the low economic consequences condition (*M* = 2.98, *SD* = 1.39, $p$ = .006). No significant differences were observed between the experimental and control conditions for *Negative Affect* (control vs. low $p$ = .610, control vs. high $p$ = .073).

In summary, intervening in the broad COVID-19 attitude network was largely successful. Results showed significant effects of interventions on the target node (except for *Economic Consequences*) and interventions affected target nodes as intended (i.e., low conditions decreased scores and high condition increased scores), serving as a manipulation check. The effect sizes of interventions were small. Interventions in the third wave, aimed at the nodes *Trust* and *Social Norm*, resulted in significant differences between all intervention and control conditions. The interventions aimed at increasing the central node *Trust* also resulted in significant effects on two connected nodes (i.e., *Measures Support* and *Social Norm*), which might indicate a causal relation. Interventions aimed at the peripheral node *Social Norm* did not affect other nodes. The intervention in the last wave aimed at the central node *Measures Support* resulted in significant differences for the target node between the experimental conditions (low and high), but not the control conditions. The interventions aimed at the rather peripheral node *Economic Consequences* did not significantly affect the target node. Regarding observed effects, it should be noted that given the naturalistic and complex setting of this study the existence of confounding variables cannot be ruled out. The next part presents results of mediation analyses aimed at examining whether significant intervention effects on nodes *connected* to target nodes can be explained by target nodes.

## Understanding intervention effects through broad attitude network structures

As mentioned, it was hypothesized that interventions aimed at target nodes (e.g., trust) would affect that target node, and that effects on nodes connected to target nodes (e.g., measures

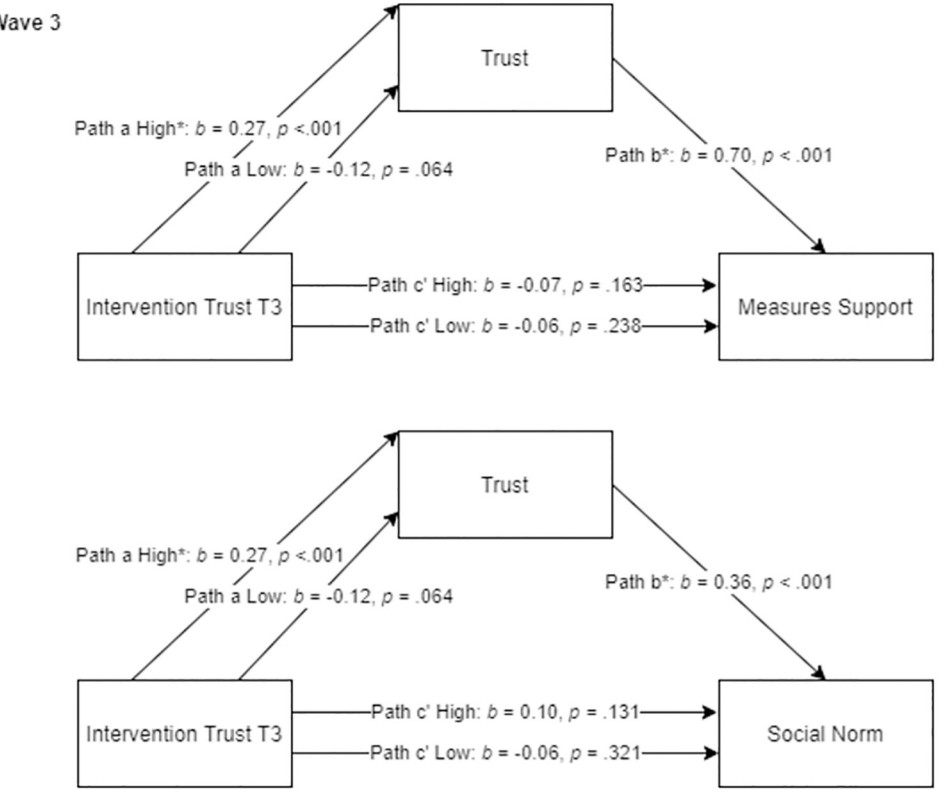

**Fig 5. Significant mediation models with nodes connected to the intervention's target node as dependent variables.** Values of indirect effects are provided in the text. * indicates significant effects (for path a and c' in reference to control group). Plots indicated that the homoscedastic assumption might be violated. Repeated analyses with transformed variables revealed comparable results.

support for trust) are induced through target nodes (e.g., interventions change trust and therefore trust changes measures support). We thus tested whether target nodes served as mediators for downstream effects of interventions. Although mediation analysis cannot provide evidence of causal mediation effects, results of such an analysis suggesting that an effect is mediated is considered a possible (first) step to investigate causal structure.

Fig 5 depicts mediation models with nodes connected to the intervention's target node as dependent variables. As described, the intervention aimed at the node *Trust* in the third wave significantly affected, in addition to the target node, the connected nodes *Measures Support* and *Social Norm*. There was a significant indirect effect of the trust intervention on the node *Measures Support* via the node *Trust* of being in the high intervention condition in reference to the control group ($ab$ = 0.19, SE = 0.05, 95% CI [0.10, 0.27]). This did not apply to the low intervention condition in reference to the control group ($ab$ = -0.08, SE = 0.05, 95% CI [-0.18, 0.01]). Similar effects were found for *Social Norm*: There was a significant indirect effect of the trust intervention on the node *Social Norm* via the node *Trust* of being in the high intervention condition in reference to the control group ($ab$ = 0.10, SE = 0.02, 95% CI [0.05, 0.14]), but this did not apply to the low intervention condition in reference to the control group ($ab$ = -0.04, SE = 0.02, 95% CI [-0.09, 0.00]). These results indicate that effects from the high *Trust* intervention on connected nodes *Measures Support* and *Social Norm* were not induced directly by the intervention but rather indirectly by effects of the intervention on *Trust*. Thus, the target node *Trust* partially mediated effects of the high trust intervention on connected nodes.

The intervention in the third wave aimed at the node *Social Norm* did not affect connected nodes, therefore no mediation analyses were conducted for this intervention. The interventions in the last wave (e.g., *Measures Support* and *Economic Consequences*) did not result in significant differences between the experimental and the control conditions (only between experimental conditions). Subsequently, mediation analyses were not conducted for interventions in the fifth wave, since multi-categorical mediation analyses uses the control group as reference group.

In summary, results showed that effects of the first set of interventions, on nodes connected to the node the intervention was aimed at, were partially mediated by the target node. This implies that downstream effects of these interventions can be partially explained by the variables the interventions were aimed at.

## Discussion

The current study examined whether intervening in a broad attitude network based on network structure a) affected the targeted nodes, b) affected nodes other than targeted nodes, and if so, whether these effects could be explained by connections with target nodes (i.e., downstream effects). An important aspect of the present study is that we developed interventions based on insight into the broad COVID-19 attitude network. Specifically, we examined whether targeting nodes that are central in the network had different effects on the network than targeting peripheral nodes. There are two main results obtained in this study into interventions.

First, the interventions affected the targeted nodes. The first set of interventions (at wave 3) induced significant changes between all conditions. The intervention aimed at the central node even led to significant change on the target node in the subsequent measure. This appears to reveal the durability of the intervention, although confounding variables cannot be excluded given the naturalistic setting. The interventions in the last measurement (wave 5) resulted in significant differences between the two intervention conditions, but not with the control condition. These less robust effects of interventions included in the last wave can possibly be explained by participants' learning effects: The first interventions were comparable to the second interventions included in the last wave, which could make participants in the second intervention less susceptible to the message because they also participated in the first interventions. Interestingly, interventions not only affected target nodes, but also had effects on other nodes in the broad COVID-19 attitude network. The next step was to examine whether these downstream effects could be explained by the broad attitude network structure.

The second result concerns using predictions derived from network theory. We investigated to what extent the impact of interventions can be explained by the structure of the broad COVID-19 attitude network. Specifically, interventions in the first wave aimed at a node central to the network had downstream effects on connected nodes that were (partially) induced through the target node. Accordingly, an intervention aimed at a less central node had no downstream effects on other nodes. These results based on the first measurement that included interventions could however not be replicated with different nodes in the last measurement. Differences in the second set of interventions for target nodes were only found between the experimental groups and not the control group.

In summary, targeted interventions that yield relatively strong effects on variables central to a broad attitude network may have downstream effects on connected nodes which can be explained by the variable the intervention was aimed at. This implies that successful interventions on central variables may be accompanied by additional effects that should be taken into account. Network structures can help to identify possible intervention outcomes. The results

discussed above only applied if the intervention was robust enough to inflict significant differences between the experimental and control conditions. Future research could aim to include more robust interventions to substantiate claims about whether the broad attitude network structure can explain effects of interventions.

Regarding the COVID-19 case study, results of the interventions suggest two possible causal effects, namely from trust in authorities responsible for managing the pandemic to 1) the degree to which people support the behavioral measures, and 2) the perceived social norm on compliance with the measures. Although we cannot rule out a direct effect of the manipulation on these nodes, we believe a causal link from trust to support for the measures and social norms is more likely because there is no clear conceptual link between the content of our manipulation and these constructs. When relating these possible causal effects to compliance with behavioral measures during the pandemic, two results are worth mentioning. First, although trust was found to be indirectly relevant for compliance through their relation with support for the measures, no strong relation between trust in authorities and compliance with behavioral measures was observed, in contrast to previous research [26]. This could imply that a relation between trust and compliance with behavioral measures is mediated by support for behavioral measures. These results are particularly interesting given the fact that trust in authorities can vary during pandemics [42]. Future research should shed further light on the relation between trust in authorities relevant to manage the pandemic, support for behavioral measures and compliance with the measures to examine if support mediates the relation between trust and compliance. Second, in contrast to previous research, including our own previous study [12], we did not observe a strong relation between social norms and compliance with behavioral measures. A possible explanation is that these survey items referred to other people in general terms and did not specify a social group people feel related to, such as family or friends [43, 44].

It should be noted that the number of interventions studied here provide relatively few causal indications compared to the size of the broad COVID-19 attitude network. However, the longitudinal design of the research that the current study was part of allows estimating networks with directed relations between variables (i.e., predictive effects). This provides causal indications for the entire broad COVID-19 attitude network as presented in Chambon, Dalege [45]. Also, as mentioned, mediation analysis cannot provide evidence of causal effects. Future research could focus on providing experimental evidence for causal effects. Furthermore, it is unknown how attrition of respondents might have affected results. Although interventions were designed based on all participants that completed the first measurement (i.e., no drop-out), the interventions were implemented in later measurements. If respondents from the recurring sample, to whom interventions were shown, are more likely to comply with behavioral measures, this might affect results. Additionally, the ecological validity of the presented broad attitude network is unknown. Although we aimed to be comprehensive by including a broad set of relevant variables that we could draw from the literature, boundaries of networks are difficult to define since the number of psychological variables that is relevant during pandemics is arguably extremely large. Future research could include additional variables that are deemed relevant based on the scientific literature to further meet real life complex attitude networks concerning pandemics. Finally, although network analysis appears to successfully provide targets via node strength, the network as a whole appears resilient against the interventions, especially in the long term. This is consistent with the size of the network; if so many nodes keep each other in check, the effect of local interventions would be expected to be small. This necessitates modesty in our expectations of behavioral interventions, but also suggests new avenues; for instance, one could think of interventions that first destabilize the connections between certain nodes and then

intervene on them or one could think of interventions that affect large parts of the network simultaneously rather than surgical interventions.

## Conclusions

Results suggest that broad attitude network structures can provide important insights for effective interventions and explaining effects of interventions. Subsequently, this is the first study to empirically show the value of understanding network structures of broad attitude networks for designing interventions and provides an informed strategy grounded in network theory. In conclusion, this research provides preliminary evidence that cross-sectional networks and strength centrality might be useful for informing social psychological interventions.

## Supporting information

**S1 File. Supporting information on procedure, analyses and results.**
(PDF)

## Author Contributions

**Conceptualization:** Monique Chambon, Jonas Dalege, Lourens J. Waldorp, Han L. J. Van der Maas, Denny Borsboom, Frenk van Harreveld.

**Data curation:** Monique Chambon.

**Formal analysis:** Monique Chambon, Jonas Dalege, Lourens J. Waldorp.

**Funding acquisition:** Frenk van Harreveld.

**Investigation:** Monique Chambon, Jonas Dalege, Lourens J. Waldorp, Han L. J. Van der Maas, Denny Borsboom, Frenk van Harreveld.

**Methodology:** Monique Chambon, Jonas Dalege, Lourens J. Waldorp, Han L. J. Van der Maas, Denny Borsboom, Frenk van Harreveld.

**Project administration:** Monique Chambon, Frenk van Harreveld.

**Supervision:** Frenk van Harreveld.

**Validation:** Monique Chambon, Jonas Dalege, Lourens J. Waldorp, Han L. J. Van der Maas, Denny Borsboom, Frenk van Harreveld.

**Visualization:** Monique Chambon.

**Writing – original draft:** Monique Chambon.

**Writing – review & editing:** Jonas Dalege, Lourens J. Waldorp, Han L. J. Van der Maas, Denny Borsboom, Frenk van Harreveld.

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
