## [Decision Letter · Decision Letter 0]

23 May 2022

PONE-D-22-06537Tailored interventions into broad attitude networks towards the COVID-19 pandemicPLOS ONE

Dear Dr. Chambon,

Thank you for submitting your manuscript to PLOS ONE. I had the pleasure of having two exert reviewers who, as you will read, provide an extremely detailed and constructive review of your manuscript. I share their positive appreciation and invite you to submit a revised version of the manuscript that addresses the points raised during the review process.Therefore, we invite you to submit a revised version of the manuscript that addresses the points raised during the review process.

We look forward to receiving your revised manuscript.

Kind regards,

Adriaan Spruyt, Ph.D

Academic Editor

PLOS ONE

Journal Requirements:

2. We note that Figure 2 in your submission contain copyrighted images. All PLOS content is published under the Creative Commons Attribution License (CC BY 4.0), which means that the manuscript, images, and Supporting Information files will be freely available online, and any third party is permitted to access, download, copy, distribute, and use these materials in any way, even commercially, with proper attribution. For more information, see our copyright guidelines: http://journals.plos.org/plosone/s/licenses-and-copyright.

Reviewers' comments:

Reviewer's Responses to Questions

**Comments to the Author**

1. Is the manuscript technically sound, and do the data support the conclusions?

Reviewer #1: Yes

Reviewer #2: Yes

2. Has the statistical analysis been performed appropriately and rigorously? 

Reviewer #1: Yes

Reviewer #2: Yes

3. Have the authors made all data underlying the findings in their manuscript fully available?

Reviewer #1: Yes

Reviewer #2: Yes

4. Is the manuscript presented in an intelligible fashion and written in standard English?

Reviewer #1: Yes

Reviewer #2: Yes

5. Review Comments to the Author

Reviewer #1: In this paper, the authors present the results of a study in which they surveyed 6093 participants regarding the covid-19 pandemic at five time points and examined structure in responding. The structure observed during the first measurement was also used to select specific factors as intervention targets based on their relation to other factors. During the third and fifth measurement, two factors were the target of a persuasion intervention and it was examined whether the interventions produced (1) changes in these factors and (2) produced changes in related factors. The former hypothesis was confirmed in most cases whereas the second hypothesis was dis-confirmed in most cases.

Overall, I found this paper very interesting to read. It reports a very extensive and well developed research project that addresses an innovative and timely research question. The question how interventions for important behavioral problems (e.g., related to the covid-19 pandemic) can be designed well is of crucial importance. Taking a network approach here and testing the value of this network approach for changing behavior related to real-world problems is quite novel and may have great potential.

The authors are very thoughtful in their explanation of the research rationale, design and results. The writing is clear and the paper presents an interesting story that is easy to follow with sufficient detail (e.g., in terms of design, results,…). The experiments were also adequately powered and well-designed and the analyses seemed suitable. The data and analysis scripts were also made available on OSF (together with a clear codebook).

That said, there are a number of things that came to mind while I was reading the paper that the authors might use to further improve their paper.

(1) Most importantly, the authors should try to more clearly separate the level of description (behavior) and the mental level of explanation (nodes). The authors often talk about effects on “nodes” but if these nodes are defined at the mental level, then such effects cannot be directly observed. The authors may use an intervention to TARGET specific nodes as defined within the framework, but they cannot know whether they INFLUENCE these nodes (or whether they exist). The authors are only postulating a relation of this node to specific questions in the survey and they can only observe responses on these questions. It would therefore be helpful if the authors clearly separate the behavioral effects and their explanation within the framework. In this way, it can also be clarified that the framework is only a tool - e.g., to build more effective treatments - rather than an accurate representation of the human mind.

(2) Related to the previous point, there was some unclarity about the research aims and, relatedly, the relevance of the research question. What were the authors' aims with this research? Was it to test the value of using the framework to influence behavior? Was it to test the value of using the framework to predict behavior? Was it to test the validity of the framework (I hope not, as that would seem impossible with this data – or with any data for that matter)? Given this aim, what was the conclusion of the research? It was a bit difficult to find out what would be the take home message after reading this paper. The authors did specify hypotheses in their introduction but the hypotheses were not clearly related to the goals nor were they as specific as one might want. How are we to interpret the results in light of these hypotheses? I guess many crucial hypotheses were not confirmed (except for the one's about the manipulation influencing the variable of interest - but I guess that is simply a matter of choosing a targeted intervention rather than a test of the framework), so does this mean that using the framework would not be as valuable as expected? Note that it is of course entirely OK if this research was very exploratory, but it would be good then if this is indicated more clearly.

(3) The authors sometimes seem to interpret their results in terms of causal relations. For instance, p. 24 : “The interventions aimed at increasing the central node Trust also resulted in significant effects on two connected nodes (i.e., Measures Support and Social Norm), which indicates a causal relation.” This seems unwarranted. It is not because you targeted change in specific questions (related to the trust Node) that the construct that these questions are assumed to probe is indeed what was changed by the intervention. It is entirely possible that the intervention produced changes in several related constructs. Indeed, evidence in this regard was found for the economic consequences interventions which only influenced responses on questions for a related construct.

The authors also seem to use mediation analyses related to the causality question but it should be noted that mediation analyses are not straightforward to make such inferences (e.g., see Agler & De Boeck, 2017) which makes it difficult to understand what these analyses add. It would be good to discuss why the authors want to use mediation analyses here (in relation to their limitations).

(4) It is unclear why the authors did not choose to engage in pre-registration (or did they?). This can be quite valuable especially for these very extensive research projects with a large number of variables and possible analyses.

(5) There is little information about statistical power. Given the high drop-out, was there still sufficient power to find between-subject effects in the fifth wave?

(6) Unclear what this sentence meant: “Finally, although network analysis appears to successfully provide targets, the network as a whole appears resilient against the interventions, especially in the long” (p.29).

Reviewer #2: The authors present what I think is a very important contribution to the applied network psychometrics literature, and I enjoyed reading it. For the editor’s information, the network literature is largely missing intervention studies testing some of the hypotheses that have come from network theory. In particular, the question of whether cross-sectional network structures can inform interventions has been an open question and arguably heavily criticised idea in recent years. In the present piece, the authors study targeted intervention on attitude networks within the context of the COVID-19 pandemic. That is, they estimate a cross-sectional network, and then design and test an intervention based on metrics that are purported to potentially capture the ‘importance’ of variables in the network. They present preliminary evidence that the network metric of strength centrality may be useful to inform intervention at the population level.

I recommend that the article is considered for publication with minor revisions, by either incorporating some changes/edits regarding some queries outlined below, or of course, whether the authors can clarify/argue some methodological points I allude to below. The rest of my review works through the paper in order.

Introduction:

• Line 78: starting from “Calculating…..” I think this is a mis-leading sentence because it implies the study is about network analysis of panel data, but no panel network models are estimated in the present article.

• Line 87: start from “Such network properties…….” To the end of the paragraph. I think this would be strengthened with a few edits to include the following points, which I think would further highlight the strengths and importance of the study:

o References missing that initially suggested the idea of the centrality hypothesis

o I think it should be acknowledged the criticisms that the centrality hypothesis has had…..

o Related to the above bullet point, to my knowledge, strength centrality as a metric to inform intervention is not

undisputed just because of direction of effects but also because of the possibility of missing latent common causes, the

boundary specification problem, and whether between subject networks are appropriate to inform intervention

strategies? If I am not incorrect, then I think this should be included, because the importance of the current paper is

that it provides evidence cross-sectional networks and strength centrality might be useful for informing interventions at

the population between subjects level.

o From line 98 when discussing Zwickers study, from my recollection, Zwicker et al, whilst intervening on a high centrality

node made no comparison to a low centrality intervention, which the present study does. I think the introduction would

be strengthened by including how this study has gone beyond what other intervention studies like Zwicker et al have in

the literature i.e. formally compared high v low centrality & tested mediation.

Methods:

• Line 174: I think the sample size information is unclear. The references cite simulations on Ising and GGM models not MGMs. If sample size was not determined a priori based on simulations of an expected MGM structure, I think this should be made clearer. (the stability of estimates as provided in the R code is useful, but in the main text what exactly the decisions around sample size were is not clear, if it was as broad as ‘collect as large a sample as possible but make sure we check stability of estimated network’, I think this should be made clear).

• Around line 197 - It is unclear to me whether subjects that participated in the 1st intervention (3rd wave) were excluded or legible for inclusion in the 2nd intervention (5th wave). I think should be made clearer and the possible implications discussed.

• Line 214 – 219 starting from “Node strength…..” a minor point, but here it defines strength centrality as the average conditional association, but then explains calculating it as simply the sum of edge weights (and not divided by the number of edge weights, so not an average?).

• The attitude network has lots of negative edge weights – was this expected? Can the authors mention why strength centrality rather than say expected influence was used?

• Line 267: I don’t know what PLOS policy is on this, but should how randomisation was performed be explained?

• Line 283 paragraph: The authors use 10-fold cross validation for selecting the tuning parameter in the regularised MGM in a large sample. I think it would be useful to justify the choice of CV for model selection (and perhaps also justify the choice of regularisation?), compared to say EBIC. Did the authors want to ‘err on the side discovery rather than caution’?

• Line 294-295 - It mentions the alpha level was set at p < 0.01 to “focus on the strongest effects”. Is this a correct interpretation of a p value? It reads as though referring to effect size, but would lowering the alpha level not just reduce the type II error rate (i.e. not necessarily focus on “strongest effects”).

• I think it would be useful for the reader to have all the information related to the intervention comparison statistical analysis in the same section rather than having to refer to footnotes

• Line 316 – Can the authors clarify here what the edge weights are in a MGM for readers unfamiliar with the model

• One thought I had here is about the validity of strength centrality in a weighted MGM. To my understanding, edges in a MGM can only be compared to edges of the same type e.g. a gaussian – gaussian edge (G – G) cannot be compared to a gaussian – binary edge (G – B) because one is an average of the same type of coefficient, but the other is an average of two different types of coefficient. In this model, you have three edge weight types: G – G, G – B, and B – B. Is strength centrality effected by whether the edges present have more or less of one type of edge weight? Assume you have two gaussian nodes, with exactly the same strength centrality, but one nodes edge set is {G-G, G-G, G-B}, and the others edge set is {G-G, G-B, G-B}, can strength centrality be validly compared? I haven’t looked at all the edge types for the intervened nodes, but is this an issue for the current study? If it’s not, then this isn’t a problem. I’m not sure what the answer is, but it was a thought I had, that maybe other readers will have so I think it should be considered here?

• Related to the MGM, the authors use edge weight comparisons and test these for significance with results supplied on the OSF link and a few results mentioned in text. Related to the above bullet point, I again wonder whether comparing all edges to each other for significance is valid in an MGM? Is it only that edges of the same type can be compared to each other for significance, and edge weights of different types cannot be compared to each other?

Results:

• Table 3 presents the means and standard deviations for all variables, but the text reports the medians. Is there a reason for this? Is one not more appropriate than the other for a given variable and should be used consistently?

• I may have missed something here but regarding the group comparisons and footnote IV: It mentions Kruskal-Wallis test was used to compare differences between the three groups but it doesn’t mention what was used for pairwise comparisons? Was this pairwise Mann Whitney U tests?

• Line 366 – this is a qualitative statement about the possible role of negative affect, should this be caveated? e.g. “possibly plays a role”, especially given bridge centrality was not calculated

• This might not be a problem when in print but in the PDF supplied the resolution of the network figure is not very good making some nodes difficult to identify.

• Line 374 – should “condition association” be “conditional association”. Same for line 388?

• Line 390 - ….”and therefore important” should probably be caveated e.g. “and therefore potentially important”.

• Line 395 – I think the evidence for “importance” is over-stated here. The edge weight connections simply suggest they may be important factors that affect compliance. Relatedly on line 397, start of second sentence, I think “Most important” should be ‘toned down’.

• This is a minor point that shouldn’t affect any conclusions, but can the authors check the strength centrality measures and perhaps clarify how values were rounded because this isn’t reported in text (unless I missed it)? For example, in text (line 438), Trust is reported as 1.80 and Social Norm as 1.00, using the data and R script provided I have Trust as 1.83 (rounded to 2dcp) and Social Norm as 1.02 (rounded to 2dcp). Likewise for Wave 5, the text (line 489) reports the wave 1 centralities of Measures Support as 2.10 and Economic Consequences as 1.10, but I have Measures Support as 2.05 and Economic Consequences as 1.09

• Line 456 – If I have read this correctly (and assuming I understand the non-parametric tests used correctly!), the text says that participants in the high trust condition scored significantly higher on Social Norm than participants in the low trust condition and control condition. But the median of social norm in high trust was 5, and the median in the control condition was 5. I could well be wrong here because I’m not a statistician, but it’s worth mentioning because other readers may have the same thought……can we conclude that participants scored higher (with a presumably Mann Whitney post-hoc test after a significant Kruskal Wallis test?) when they have the same median? Does this not just imply the distributions are different if the median is the same? – can we conclude the direction of distributional differences is in the favour of the high trust group? (Same for line 466 to 468).

• Intervention wave 5 – with the measures support intervention, there is a significant effect on social norm but this is not mentioned in the text (just the table), I think this should be added to the text as well so it contains the overall picture of the intervention effects.

• Intervention wave 5 from line 503 – There was a significant effect on negative affect in this intervention, I think this should also be mentioned in the text for same reason as previous bullet point.

• I don’t know enough about formal mediation analysis to comment on this, but I do think this is a particular strength of the paper that mediation analysis was conducted to test whether the effects could be explained via mediation of the target node

Discussion:

• Line 632 (starting from the “longitudinal design…”) to line 634. I think this is just a wording issue but the sentence talks about the current design enabling estimating directed networks from the panel data structure. It references a study in which this was done, but I think this is mis-leading because it reads as though this is what was done in the current study. Perhaps consider rewording?

• Line 650: “….interventions, especially in the long” – missing word, ‘run’?

Miscellaneous:

• I was able to reproduce the network structure with the code and data supplied, but the R script didn’t contain the intervention comparisons which it states was conducted in SPSS

• The edge weight accuracy, and edge and centrality difference test figures are missing from the supplement and instead are provided on the OSF. I think the edge weight accuracy/stability plot should be provided in the supplement.

• Does the R script provided also require loading of the psych library for the fisherz function on line 586? (I had to load the psych library to get this to work).

6. PLOS authors have the option to publish the peer review history of their article (what does this mean?). If published, this will include your full peer review and any attached files.

Reviewer #1: **Yes: **Pieter Van Dessel

Reviewer #2: No

---

## [Author Response · Author response to Decision Letter 0]

14 Jul 2022

Dear Dr. Spruyt and reviewers,

Please find enclosed a revised version of our manuscript ‘Tailored interventions into broad attitude networks towards the COVID-19 pandemic’ (PONE-D-22-06537). We would like to thank you for the opportunity for an improvement of the manuscript. We found the feedback of the reviewers very helpful. We have tried to address all the issues raised by the reviewers and incorporated them in the present version of the manuscript. We will address each issue more specifically below. 

Comments by editors

Ensure that the manuscript meets PLOS ONE's style requirements, including those for file naming

We adjusted the manuscript and file naming in line with PLOS ONE's style requirements. 

Comply with copyright policy of images

We removed all images to comply with the copyright policy. 

Comments by reviewers

First, we were delighted with the reviewers’ positive evaluation of our research, and we are grateful for their feedback that was very helpful in improving the manuscript. Next, we will turn to the more specific comments made by each of the reviewers:

Reviewer #1

- The reviewer recommends to more clearly separate the level of description (behavior) and the mental level of explanation (nodes).

Agreeing with the reviewer that this is a very important distinction, we tried to make this distinction clearer in the revised manuscript. We now write in the introduction: 

“Networks can be calculated with (e.g., survey) data. Nodes represent the measured psychological constructs, which can consist of single items or a combination of items (e.g., average on multiple items). Edges between nodes in psychological networks cannot be directly observed and are therefore parameters that are estimated from data (Epskamp et al., 2018).”

Also, we changed the wording ‘influencing nodes’ into ‘affecting nodes’. Finally, we included the following footnote in the introduction:

“Note that the wording of affecting nodes is used for brevity and refers to affecting scores on the item(s) forming that node. States and changes of nodes were only measured at the behavioral level. Our research can therefore only speak to that level of explanation and is not on the level of mental processes.”

- Reviewer 1 points out that there was some unclarity about the research aims and, relatedly, the relevance of the research question.

We agree with the reviewer that the research aim and the take home message could be stated more clearly. At the end of the first paragraph in the revised manuscript we now write:

“In this study, we empirically explore how the network perspective can inform social psychological interventions. The aim of the current research is to investigate how broad attitude networks respond to tailored interventions aimed at variables that differ in their connectiveness with other variables.”

Also, we added the following sentence as the last sentence in the conclusion (a statement that was suggested by the second reviewer):

“In conclusion, this research provides preliminary evidence on that cross-sectional networks and strength centrality might be useful for informing social psychological interventions.”

The relevance of the research question is substantiated in the revised manuscript by including the criticism the centrality hypothesis had, as also suggested by Reviewer 2. This highlights the importance of empirical research for the debate around the centrality hypothesis. 

- The reviewer notices that results are sometimes interpreted in terms of causal relations where this seems unwarranted. Targeting items that are related to a node does not mean that the construct that these items are assumed to probe is indeed what was changed by the intervention, because it is entirely possible that the intervention produced changes in several related constructs.

We agree with the reviewer that this requires more nuance. Although we cannot rule out a direct influence of the manipulation on these nodes, we believe a causal link between the Trust node and these latter two nodes is more likely because there is no clear conceptual link between the content of our manipulation and these nodes. We now write this in the discussion of the revised manuscript. Furthermore, we attenuated interpretations of causal relations throughout the manuscript in terms of that results might indicate causal relations. 

- Reviewer 1 expresses concerns about using mediation analyses to make inferences about causality and recommends to discuss why mediation analyses is used. 

Agreeing with the reviewer that formal mediation inferences cannot be made based on mediation analysis alone, we now write in the revised manuscript: “Although mediation analysis cannot provide evidence of causal mediation effects, results of such an analysis suggesting that an effect is mediated is considered a possible (first) step to investigate causal structure.”

In the discussion we now write in the limitation section: “Also, as mentioned, mediation analysis cannot provide evidence of causal effects. Future research could focus on providing experimental evidence for causal effects.”

- The reviewer mentions that it is unclear whether the study was pre-registered and the reason for this.

Although we agree that pre-registration is a valuable route, this research was not preregistered due to its explorative nature. We now state this in the method section of the manuscript.

- Reviewer 1 points out that there is little information about statistical power and asks whether there was still sufficient power to find between-subject effects in the fifth wave?

A total of 2399 participants completed the fifth wave, which is considered to provide sufficient power. This sample size far exceeds the minimal number of participants necessary to detect between-subject effects. 

We now write in the methods section of the revised manuscript:

“Regarding sample size, the aim was to collect as large a sample as possible to ensure sufficient power to find between-subject effects in the last measurement, after which we checked the stability of the estimated network.”

- The reviewer comments that it is unclear what this sentence means: “Finally, although network analysis appears to successfully provide targets, the network as a whole appears resilient against the interventions, especially in the long” (p.29).

The second part of the sentence is now clarified, and the missing word in the end of the sentence is added. We now write in the revised manuscript:

“Finally, although network analysis appears to successfully provide targets via node strength, the network as a whole appears resilient against the interventions, especially in the long term.”

Reviewer #2

For each comment of Reviewer 2, we first present the comment (verbatim) in bold font, and our reply in regular font (with manuscript citations in italics font).

Introduction:

• Line 78: starting from “Calculating…..” I think this is a mis-leading sentence because it implies the study is about network analysis of panel data, but no panel network models are estimated in the present article.

We agree that this might cause confusion about the design of this research and therefore removed the last part of the sentence (i.e., ‘(…), which is what we employ in the present research’ is deleted).

• Line 87: start from “Such network properties…….” To the end of the paragraph. I think this would be strengthened with a few edits to include the following points, which I think would further highlight the strengths and importance of the study:

o References missing that initially suggested the idea of the centrality hypothesis

The revised manuscript now contains the reference to Borsboom & Cramer (2013) in which this idea was initially proposed.

o I think it should be acknowledged the criticisms that the centrality hypothesis has had…..

o Related to the above bullet point, to my knowledge, strength centrality as a metric to inform intervention is not undisputed just because of direction of effects but also because of the possibility of missing latent common causes, the boundary specification problem, and whether between subject networks are appropriate to inform intervention strategies? If I am not incorrect, then I think this should be included, because the importance of the current paper is that it provides evidence cross-sectional networks and strength centrality might be useful for informing interventions at the population between subjects level.

Agreeing with the added value of discussing criticisms that the centrality hypothesis has had, we now briefly discuss this and refer readers to two excellent papers that provide an overview of such criticisms (including relevant references). In the revised manuscript we now write:

“Theoretically, changing a central (i.e., highly connected) node is likely to have a more profound effect on a network than changing a peripheral (i.e., less connected) node, given the central nodes’ relatively high connectiveness to other nodes (Borsboom & Cramer, 2013). This is however not undisputed in undirected networks for reasons such as missing latent common causes and problems regarding the specification of boundaries of networks (see Bringmann et al., 2019; Hallquist et al., 2021 for an overview). Also, high connectivity of a node can result from different scenario’s, namely a) that the node highly affects other nodes, b) that the node is highly affected by other nodes, or c) a combination of the first two directions of effects. Results of interventions depend on these directions of effects: intervening on a node that highly affects other nodes is likely to have a profound effect on the network, whereas changing a node that is highly affected by other nodes is unlikely to (durably) affect the network.”

In the conclusions section of the revised manuscript we now write:

“In conclusion, this research provides preliminary evidence that cross-sectional networks and strength centrality might be useful for informing social psychological interventions.”

o From line 98 when discussing Zwickers study, from my recollection, Zwicker et al, whilst intervening on a high centrality node made no comparison to a low centrality intervention, which the present study does. I think the introduction would be strengthened by including how this study has gone beyond what other intervention studies like Zwicker et al have in the literature i.e. formally compared high v low centrality & tested mediation.

We are glad with this suggestion and agree that this would strengthen the introduction. In the revised manuscript we now write in the section on the present research: 

“The current study thus goes beyond what earlier intervention studies have done (e.g., Zwicker et al., 2020), by formally comparing intervention effects on nodes with relatively high and low centrality, and testing mediation effects.”

Methods:

• Line 174: I think the sample size information is unclear. The references cite simulations on Ising and GGM models not MGMs. If sample size was not determined a priori based on simulations of an expected MGM structure, I think this should be made clearer. (the stability of estimates as provided in the R code is useful, but in the main text what exactly the decisions around sample size were is not clear, if it was as broad as ‘collect as large a sample as possible but make sure we check stability of estimated network’, I think this should be made clear).

Clarification would indeed benefit the readers’ understanding and we adjusted the manuscript accordingly. We now write in the method section:

“Regarding sample size, the aim was to collect as large a sample as possible to ensure sufficient power to find between-subject effects in the last measurement, after which we checked the stability of the estimated network. We aimed for, and far exceeded, a minimum of 500 participants because this is the highest the advised number of participants for a moderately sized network with either continuous or binary data (Epskamp, 2017; van Borkulo et al., 2014), and these types of data were combined in the current study.”

• Around line 197 - It is unclear to me whether subjects that participated in the 1st intervention (3rd wave) were excluded or legible for inclusion in the 2nd intervention (5th wave). I think should be made clearer and the possible implications discussed.

We inserted clarification in the methods section of the revised manuscript:

“Respondents that participated in the first intervention (third wave) were also eligible for inclusion in the second intervention (fifth wave). Therefore, respondents that were included in the second intervention were also included in the first intervention.”

In the discussion we now write:

“These less robust effects of interventions included in the last wave can possibly be explained by participants’ learning effects: The first interventions were comparable to the second interventions included in the last wave, which could make participants in the second intervention less susceptible to the message because they also participated in the first interventions.”

• Line 214 – 219 starting from “Node strength…..” a minor point, but here it defines strength centrality as the average conditional association, but then explains calculating it as simply the sum of edge weights (and not divided by the number of edge weights, so not an average?).

We now see how this might cause confusion and thus removed the word average in this context.

• The attitude network has lots of negative edge weights – was this expected? 

Given the explorative nature of this empirical, data-driven study, we did not have hypotheses about specific edges. Nevertheless, given that we did not only include variables with non-arbitrary coding (e.g., when high values of all variables indicate more psychopathology; Borsboom et al., 2021), negative relations were to be expected.

• Can the authors mention why strength centrality rather than say expected influence was used?

Strength, Closeness and Betweenness are most commonly used. However, as mentioned by Borsboom et al. (2021), closeness and betweenness can be problematic since they treat association as distances. Expected influence, which includes signs of edge weights, can be appropriate for variables with non-arbitrary coding (see previous comment), which was not the case in this research. 

The revised manuscript now includes the following text in the introduction:

“The most commonly used centrality measures are Strength (calculated by summing absolute edge weights), Closeness (calculating distances between nodes based on the shortest path length) and Betweenness (calculated based on how often it lies on the shortest path between nodes) of nodes (Borsboom et al., 2021). Of these centrality metrics, Closeness and Betweenness are considered least suitable for psychological networks (Bringmann et al., 2019), and can be problematic because they treat associations between nodes as distances (Borsboom et al., 2021).”

In the method section we now mention: 

“(..) node Strength is considered the most suitable centrality measure for psychological networks. This research thus focuses on Strength as a centrality measure.”

• Line 267: I don’t know what PLOS policy is on this, but should how randomisation was performed be explained?

In the revised manuscript we mention that randomization was conducted by the software in which the questionnaire was programmed (i.e., Qualtrics).

• Line 283 paragraph: The authors use 10-fold cross validation for selecting the tuning parameter in the regularised MGM in a large sample. I think it would be useful to justify the choice of CV for model selection (and perhaps also justify the choice of regularisation?), compared to say EBIC. Did the authors want to ‘err on the side discovery rather than caution’?

We have opted for k-fold cross validation (CV) because it emphasizes prediction more than with the EBIC, but often they give similar results. Both methods have also been investigated and have good properties. The k-fold CV is sometimes more conservative (i.e., allows fewer edges) than the EBIC, especially for smaller sample sizes. We included this in a footnote in the revised manuscript.

• Line 294-295 - It mentions the alpha level was set at p < 0.01 to “focus on the strongest effects”. Is this a correct interpretation of a p value? It reads as though referring to effect size, but would lowering the alpha level not just reduce the type II error rate (i.e. not necessarily focus on “strongest effects”).

We fully agree with the reviewer that this should be interpreted in terms of reducing the probability of error rates. We adjusted the alpha level to reduce the type 1 error probability and adjusted this in the revised manuscript accordingly. 

• I think it would be useful for the reader to have all the information related to the intervention comparison statistical analysis in the same section rather than having to refer to footnotes.

In the revised manuscript this information is provided in the main text. 

• Line 316 – Can the authors clarify here what the edge weights are in a MGM for readers unfamiliar with the model.

The revised manuscript now includes the following description: “Edge weights are regression coefficients that represent the strength of a relation between two nodes after removing effects from all other nodes in the network.”

• One thought I had here is about the validity of strength centrality in a weighted MGM. To my understanding, edges in a MGM can only be compared to edges of the same type e.g. a gaussian – gaussian edge (G – G) cannot be compared to a gaussian – binary edge (G – B) because one is an average of the same type of coefficient, but the other is an average of two different types of coefficient. In this model, you have three edge weight types: G – G, G – B, and B – B. Is strength centrality effected by whether the edges present have more or less of one type of edge weight? Assume you have two gaussian nodes, with exactly the same strength centrality, but one nodes edge set is {G-G, G-G, G-B}, and the others edge set is {G-G, G-B, G-B}, can strength centrality be validly compared? I haven’t looked at all the edge types for the intervened nodes, but is this an issue for the current study? If it’s not, then this isn’t a problem. I’m not sure what the answer is, but it was a thought I had, that maybe other readers will have so I think it should be considered here?

In modeling the binary (Betnoulli) and continuous (Gaussian) random variables we used the parameters for each of the possible pairwise interactions have the same interpretation. This is because in the conditional probability for these distributions from an exponential family, we obtain coefficients for the sufficient statistics for the pairwise interactions (see Haslbeck & Waldorp, 2020, equation 5 and below). This results in the parameters having similar interpretations. If we were not using distributions from the exponential family, we would not be able to do this.

• Related to the MGM, the authors use edge weight comparisons and test these for significance with results supplied on the OSF link and a few results mentioned in text. Related to the above bullet point, I again wonder whether comparing all edges to each other for significance is valid in an MGM? Is it only that edges of the same type can be compared to each other for significance, and edge weights of different types cannot be compared to each other?

In terms of interpretation of the parameters, this works because of the previous comment. Additionally, the difference between edge weights is obtained with a bootstrap distribution. This assumes that the estimates have been obtained with a similar procedure each time, which was the case.

In the section of the supplement that presents the edge weight comparisons and tests we now also state “These analyses are conducted with bootstrap network estimation. In doing so, we specified the method to use as mgm (i.e., package bootnet, function bootnet, argument default = "mgm").”

Results:

• Table 3 presents the means and standard deviations for all variables, but the text reports the medians. Is there a reason for this? Is one not more appropriate than the other for a given variable and should be used consistently?

To provide all information relevant for the interpretation of the results we included the medians in Table 3 in the revised manuscript. The supplement now contains an extra table with the medians of every variable in the network.

• I may have missed something here but regarding the group comparisons and footnote IV: It mentions Kruskal-Wallis test was used to compare differences between the three groups but it doesn’t mention what was used for pairwise comparisons? Was this pairwise Mann Whitney U tests?

Kruskall-Wallis post hoc pairwise comparison was conducted with the Dunn-Bonferroni post hoc method (SPSS default). We now mention this in the revised manuscript.

• Line 366 – this is a qualitative statement about the possible role of negative affect, should this be caveated? e.g. “possibly plays a role”, especially given bridge centrality was not calculated.

The revised manuscript is reframed in such a way that its main focus is on compliance. That is, variables related to (mental) health are positioned in line with the other variables in the network, namely to examine their effect on compliance, and not as a separate focus as was the case with the previous version of the manuscript. We therefore removed statements on how compliance and well-being are connected, including the statement to which this comment refers.. 

• This might not be a problem when in print but in the PDF supplied the resolution of the network figure is not very good making some nodes difficult to identify.

We thank the reviewer for mentioning this and will be alert on the quality of the figures at the manuscript proof. 

• Line 374 – should “condition association” be “conditional association”. Same for line 388?

We are thankful that the reviewer noticed this and adjusted it.

• Line 390 - ….”and therefore important” should probably be caveated e.g. “and therefore potentially important”.

We agree that nuance is in place and changed it accordingly.

• Line 395 – I think the evidence for “importance” is over-stated here. The edge weight connections simply suggest they may be important factors that affect compliance. Relatedly on line 397, start of second sentence, I think “Most important” should be ‘toned down’.

As with the previous comment, we agree that nuance is important here. In hindsight we think that this summary is redundant and thus removed it from the manuscript.

• This is a minor point that shouldn’t affect any conclusions, but can the authors check the strength centrality measures and perhaps clarify how values were rounded because this isn’t reported in text (unless I missed it)? For example, in text (line 438), Trust is reported as 1.80 and Social Norm as 1.00, using the data and R script provided I have Trust as 1.83 (rounded to 2dcp) and Social Norm as 1.02 (rounded to 2dcp). Likewise for Wave 5, the text (line 489) reports the wave 1 centralities of Measures Support as 2.10 and Economic Consequences as 1.10, but I have Measures Support as 2.05 and Economic Consequences as 1.09.

The reported strength values were derived from the bootstrap centrality difference results (see supplement, node strength is presented in the diagonal boxes) instead of the network model from the mgm analysis. K-fold CV can result in minor differences due to randomness in the subsets that are cross validated. To avoid confusion we now report in the revised manuscript the strength measures from the mgm network model, similar to those reported by the reviewer, and include a table with the complete strength values in the supplement. 

• Line 456 – If I have read this correctly (and assuming I understand the non-parametric tests used correctly!), the text says that participants in the high trust condition scored significantly higher on Social Norm than participants in the low trust condition and control condition. But the median of social norm in high trust was 5, and the median in the control condition was 5. I could well be wrong here because I’m not a statistician, but it’s worth mentioning because other readers may have the same thought……can we conclude that participants scored higher (with a presumably Mann Whitney post-hoc test after a significant Kruskal Wallis test?) when they have the same median? Does this not just imply the distributions are different if the median is the same? – can we conclude the direction of distributional differences is in the favour of the high trust group? (Same for line 466 to 468).

In order to clarify these directions we now report in the revised manuscript the mean ranks for significant differences between conditions with similar medians.

• Intervention wave 5 – with the measures support intervention, there is a significant effect on social norm but this is not mentioned in the text (just the table), I think this should be added to the text as well so it contains the overall picture of the intervention effects.

This effect was mentioned in the footnote, but we now included this in the main text.

• Intervention wave 5 from line 503 – There was a significant effect on negative affect in this intervention, I think this should also be mentioned in the text for same reason as previous bullet point.

This effect was mentioned in the footnote, but we now included this in the main text.

• I don’t know enough about formal mediation analysis to comment on this, but I do think this is a particular strength of the paper that mediation analysis was conducted to test whether the effects could be explained via mediation of the target node.

We are glad that the reviewer sees this as a particular strength.

Discussion:

• Line 632 (starting from the “longitudinal design…”) to line 634. I think this is just a wording issue but the sentence talks about the current design enabling estimating directed networks from the panel data structure. It references a study in which this was done, but I think this is mis-leading because it reads as though this is what was done in the current study. Perhaps consider rewording?

We agree that this wording might cause confusion and rephrased the sentence to clarify that the current study was part of a longitudinal study. In the revised manuscript we now write: “(…) the longitudinal design of the research that the current study was part of allows estimating networks with directed relations between variables (i.e., predictive effects). This provides causal indications for the entire broad COVID-19 attitude network as presented in Chambon et al. (2021)”.

• Line 650: “….interventions, especially in the long” – missing word, ‘run’?

This is now corrected.

Miscellaneous:

• I was able to reproduce the network structure with the code and data supplied, but the R script didn’t contain the intervention comparisons which it states was conducted in SPSS

The network- and mediation analyses were conducted in R because of the required packages. All other analyses were conducted in SPSS. The R code and SPSS syntax are both provided on OSF.

• The edge weight accuracy, and edge and centrality difference test figures are missing from the supplement and instead are provided on the OSF. I think the edge weight accuracy/stability plot should be provided in the supplement.

We now included the edge weight accuracy and centrality stability plots in the supplement. The output of the difference tests is provided on OSF as PDF’s because they contain too much details to be readable in Word. This also applies to the edge weight accuracy plot. Nevertheless, we included it in the supplement to give a general impression of the width of the edge weight confidence intervals. The PDF’s of these analyses remain available via OSF, and in the supplement we refer to these documents for detailed results.

• Does the R script provided also require loading of the psych library for the fisherz function on line 586? (I had to load the psych library to get this to work).

We are glad the reviewer noticed this. This library is now added to the R script.

References: 

Borsboom, D., et al. (2021). Network analysis of multivariate data in psychological science. Nature Reviews Methods Primers. 1(1): 58.

Haslbeck, J. M. B. & L. J. Waldorp (2020). mgm: Estimating Time-Varying Mixed Graphical Models in High-Dimensional Data. Journal of Statistical Software 93(8): 1-46

Again, we are thankful for the many insightful comments made by the reviewers. Hopefully, we have adequately dealt with all of them and we are looking forward to hearing from you.

---

## [Decision Letter · Decision Letter 1]

1 Sep 2022

PONE-D-22-06537R1Tailored interventions into broad attitude networks towards the COVID-19 pandemicPLOS ONE

Dear Dr. Chambon,

Thank you for submitting your manuscript to PLOS ONE. As you will see, both reviewers are extremely satisfied with the changes made. I share their assessment. Still, Reviewer 2 sees some room for further improvement. I have therefore made the decision to formally request a "minor revision". I am convinced, however, that you can make these last adjustments quickly and easily. I look forward to receiving the updated version soon.

We look forward to receiving your revised manuscript.

Kind regards,

Adriaan Spruyt, Ph.D

Academic Editor

PLOS ONE

Journal Requirements:

Reviewers' comments:

Reviewer's Responses to Questions

**Comments to the Author**

1. If the authors have adequately addressed your comments raised in a previous round of review and you feel that this manuscript is now acceptable for publication, you may indicate that here to bypass the “Comments to the Author” section, enter your conflict of interest statement in the “Confidential to Editor” section, and submit your "Accept" recommendation.

Reviewer #1: All comments have been addressed

Reviewer #2: All comments have been addressed

2. Is the manuscript technically sound, and do the data support the conclusions?

Reviewer #1: Yes

Reviewer #2: Yes

3. Has the statistical analysis been performed appropriately and rigorously? 

Reviewer #1: Yes

Reviewer #2: Yes

4. Have the authors made all data underlying the findings in their manuscript fully available?

Reviewer #1: Yes

Reviewer #2: Yes

5. Is the manuscript presented in an intelligible fashion and written in standard English?

Reviewer #1: Yes

Reviewer #2: Yes

6. Review Comments to the Author

Reviewer #1: I thank the authors for their extensive revisions. In my opinion, the authors have decidedly improved the paper. All critical points have been addressed and I applaud the changes that were made.

Reviewer #2: Thank you for re-submitting the paper I was interested to see this again. I’ve read the replies to the reviewer comments, and I have just a few comments below which are either minor or are points that I think are appropriate to discuss post-publication.

1) Should Zeal and Neal (2021) be cited when mentioning criticism of centrality, specifically re: the boundary specification problem (doi: 10.1037/met0000426). If this paper is covered in the cited Hallquist paper cited then perhaps not, but otherwise I think Zeal probably should be cited here.

2) Should the IQR range be added to Table 3 for the medians?

3) Regarding my initial comment about whether a nodes edge set may influence strength centrality, I’m not sure the authors reply did completely address this. I understand that edges in an MGM all have the same substantive interpretation e.g. conditional associations, but what I don’t think the reply did address (unless I’m mistaken), is what effect averaging different types of regression coefficients may have on calculating strength centrality. For example (if I understand the MGM correctly), if you have a continuous node connected to two other continuous nodes then edge weights are averaged linear regression coefficients and strength centrality is the sum of these two averages. But if you have a continuous node connected to one other continuous node as well as a binary node, then one edge is an average of two linear regression coefficients, and the other edge is an average of a linear regression coefficient and a logistic regression coefficient, and strength centrality is thus the sum of different types of coefficient averages compared to the first node. Is it valid to then compare the two nodes on strength centrality? (even if the edges themselves can all still be interpreted as conditional associations)? It is however quite possible I am misunderstanding, because I am not an expert in psychometrics or statistics.

Overall, I’m satisfied that the authors addressed all points sufficiently, and as such, my recommendation would be for the paper to be published.

7. PLOS authors have the option to publish the peer review history of their article (what does this mean?). If published, this will include your full peer review and any attached files.

Reviewer #1: No

Reviewer #2: No

---

## [Author Response · Author response to Decision Letter 1]

4 Oct 2022

Subject: revision PONE-D-22-06537R1

Dear Dr. Spruyt,

Please find enclosed a revised version of our manuscript ‘Tailored interventions into broad attitude networks towards the COVID-19 pandemic’ (PONE-D-22-06537R1). We would like to thank you for the opportunity for an improvement of the manuscript. We will address each issue more specifically below. 

Comments by reviewers

1) Should Zeal and Neal (2021) be cited when mentioning criticism of centrality, specifically re: the boundary specification problem (doi: 10.1037/met0000426). If this paper is covered in the cited Hallquist paper cited then perhaps not, but otherwise I think Zeal probably should be cited here.

We agree that this paper is relevant and included the citation in the revised manuscript. 

2) Should the IQR range be added to Table 3 for the medians?

We are thankful for this suggestion and included the IQR in Table 3 in the revised manuscript.

3) Regarding my initial comment about whether a nodes edge set may influence strength centrality, I’m not sure the authors reply did completely address this. I understand that edges in an MGM all have the same substantive interpretation e.g. conditional associations, but what I don’t think the reply did address (unless I’m mistaken), is what effect averaging different types of regression coefficients may have on calculating strength centrality. For example (if I understand the MGM correctly), if you have a continuous node connected to two other continuous nodes then edge weights are averaged linear regression coefficients and strength centrality is the sum of these two averages. But if you have a continuous node connected to one other continuous node as well as a binary node, then one edge is an average of two linear regression coefficients, and the other edge is an average of a linear regression coefficient and a logistic regression coefficient, and strength centrality is thus the sum of different types of coefficient averages compared to the first node. Is it valid to then compare the two nodes on strength centrality? (even if the edges themselves can all still be interpreted as conditional associations)? It is however quite possible I am misunderstanding, because I am not an expert in psychometrics or statistics.

This is indeed an interesting and very relevant question. In general your intuition is correct, that parameters obtained from random variables with all kinds of distributions are difficult to compare. For our specific situation though, parameters are comparable. We will try and explain this better.

You could think of the following situation. Suppose we have a Gaussian random variable Y with two predictors X and Z, and X is Gaussian and Z is binary. Then changing X one unit (on a continuous scale) results in a change in Y, and this is similar to changing Z one unit (on a binary scale) resulting in a change in Y. The coefficients for X and Z are then comparable because X and Z change Y in a similar way and X and Z are exponential family random variables. So, in general such comparisons of edge parameters is not possible. But in our case they are.

We hope this answers the question.

We are again thankful for the comments made by the reviewers. Hopefully, we have adequately dealt with all of them and we are looking forward to hearing from you.

---

## [Editor Report · Decision Letter 2]

7 Oct 2022

Tailored interventions into broad attitude networks towards the COVID-19 pandemic

PONE-D-22-06537R2

Dear Dr. Chambon,

We’re pleased to inform you that your manuscript has been judged scientifically suitable for publication and will be formally accepted for publication once it meets all outstanding technical requirements.

Kind regards,

Adriaan Spruyt, Ph.D

Academic Editor

PLOS ONE

---

## [Editor Report · Acceptance letter]

18 Oct 2022

PONE-D-22-06537R2 

Tailored interventions into broad attitude networks towards the COVID-19 pandemic 

Dear Dr. Chambon:

I'm pleased to inform you that your manuscript has been deemed suitable for publication in PLOS ONE. Congratulations! Your manuscript is now with our production department. 

Kind regards, 

on behalf of

Dr. Adriaan Spruyt 

Academic Editor

PLOS ONE